# Policies Permitting LLM Use for Polishing Peer Reviews Are Currently Not Enforceable

**Rounak Saha** [1]  **Gurusha Juneja** [*2]  **Dayita Chaudhuri** [*1]  **Naveeja Sajeevan** [*1]  **Nihar B Shah** [3]  **Danish Pruthi** [1]

## Abstract

A number of scientific conferences and journals have recently enacted policies that prohibit LLM usage by peer reviewers, except for polishing, paraphrasing, and grammar correction of otherwise human-written reviews. But, are these policies enforceable? To answer this question, we assemble a dataset of peer reviews simulating multiple levels of human-AI collaboration, and evaluate five state-of-the-art detectors, including two commercial systems. Our analysis shows that all detectors misclassify a non-trivial fraction of LLM-polished reviews as AI-generated, thereby risking false accusations of academic misconduct. We further investigate whether peer-review-specific signals, including access to the paper manuscript and the constrained domain of scientific writing, can be leveraged to improve detection. While incorporating such signals yields measurable gains in some settings, we identify limitations in each approach and find that none meets the accuracy standards required for identifying AI use in peer reviews. Importantly, our results suggest that recent public estimates of AI use in peer reviews through the use of current AI-text detectors should be interpreted with caution, as they misclassify mixed reviews (collaborative human-AI outputs) as fully AI generated, potentially overstating the extent of policy violations.

 FLAIR-IISc/ai-in-peer-review

## 1. Introduction

Scientific journal editors and conference program chairs routinely rely on experts to evaluate submitted manuscripts through peer review. However, there have been growing concerns lately about reviewers delegating this task to AI systems, prompting journals and conferences to adopt a range of policies governing AI use in the review process (ICML 2025; Wiley, 2025). These concerns have also been a part of public discourse. A notable recent example involves the ICLR 2026 conference, whose reviews are publicly released. An AI-text detection company claimed that 21% of submitted reviews were AI-generated (Pangram Labs, 2025b). This claim attracted widespread attention, and was discussed and reported (Naddaf, 2025) with little scrutiny.

Two types of policies are prominently adopted by conferences and journals. One policy is to completely prohibit use of LLMs by reviewers (e.g., by ICML 2025), and we will refer to this as **No-LLM-use policy**. A second common policy is to allow reviewers to use LLMs only for paraphrasing and grammar correction of their reviews, while the core arguments and content of the review must be human-written. Such a policy is currently adopted widely, e.g., by ICLR 2026, ACL Rolling Review, Taylor & Francis, and many others. We collectively call these policies as **Polishing-only policies**. For instance, EMNLP 2025, citing the ACL Policy on Publication Ethics, states:

> *"it is acceptable to use LLMs for paraphrasing, grammatical checks and proof-reading, but not for the content of the (meta-)reviews."*
>
> Reviewer Policies, EMNLP 2025

With such policies precisely delineating acceptable LLM use, and detectors claiming to be capable of screening tens of thousands of reviews overnight, it might be tempting to conclude that the machinery for enforcing these policies is already in place. Enforcement of policies, however, hinges critically on the key question: *To what extent can the extent of AI use in peer reviews be detected?* To put things into perspective, at the scale of NeurIPS 2025 with over 75,000 reviews, even a $0.1\%$ false positive rate can translate to 75 wrongful accusations of academic misconduct. A $3\%$ false positive rate can result in an alarmingly high 2,250 wrongful accusations. Therefore, any credible enforcement regime would require a far lower false positive rate. Yet, the reliability of these detectors in detecting AI use in peer reviews remains largely unexamined. In this work, we directly in-

---

[*]Equal contribution  [1]Indian Institute of Science [2]University of California, Santa Barbara [3]Carnegie Mellon University. Correspondence to: Rounak Saha <rounaksaha@iisc.ac.in>, Danish Pruthi <danishp@iisc.ac.in>.

*Proceedings of the $43^{rd}$ International Conference on Machine Learning*, Seoul, South Korea. PMLR 306, 2026. Copyright 2026 by the author(s).

vestigate this question by evaluating state-of-the-art AI text detectors on a curated dataset of reviews spanning multiple levels of human-AI collaboration.

**Data.** We curate a dataset of 50, 156 peer reviews spanning multiple levels of human-AI collaboration. We start with **human**-written reviews from the pre-ChatGPT-era ML conferences, sourced from the PeerRead dataset (Kang et al., 2018). We then use popular LLMs to generate synthetic reviews for the same papers to simulate the following levels of AI-assistance:

(1) *AI-generated with Basic Prompts* (**AI-BP**): Reviews generated by prompting an LLM with the paper and conference reviewing guidelines;

(2) *AI-generated with Elaborate Prompts* (**AI-EP**): Reviews generated by prompting with additional tips apart from paper and official reviewing guidelines;

(3) *AI-generated with Human Input* (**AI-HI**): Reviews generated by an LLM by expanding on key assessment points extracted from a human-written review, simulating minimal human input;

(4) *Human-written AI-polished* (**H-AI**): Human-written reviews polished by an LLM for grammar, flow and clarity, the only form of LLM use permitted under Polishing-only policies; and

(5) ***Humanized AI-BP and H-AI*** reviews, adversarially paraphrased to evade AI detection.

**Detectors evaluated.** We evaluate five AI text detectors, comprising three open-source and zero-shot AI detectors, namely LogLikelihood (Solaiman et al., 2019), Fast-DetectGPT (Bao et al., 2024), and Binoculars (Hans et al., 2024), which output scalar detection scores, as well as two commercial classifiers, Pangram (Emi & Spero, 2024; Thai et al., 2025) and GPTZero (Tian & Cui, 2026), which produce multi-class categorical predictions ("AI", "Mixed", "Human"). For the zero-shot detectors, we generate binary predictions by thresholding their output scores. The thresholds are chosen to achieve a hard $0\%$ false positive rate on human-written reviews in a calibration set.

**Potential peer-review-specific advantages.** As opposed to general AI-text detection, the application of detecting AI use in peer reviews presents unique opportunities. For instance, the manuscript that is under review is likely a part of the prompt used to generate the review. Most open-source detectors like LogLikelihood, DetectGPT, Fast-DetectGPT and Binoculars rely on perplexity or surprisal-based metrics, which can be naturally extended to incorporate paper context by computing the likelihood of a review conditioned on the manuscript. Another way to incorporate paper context is to first generate a set of LLM-written reference reviews for the manuscript, and then use similarity between a candidate review and these references as a detection signal (Yu et al., 2025). Secondly, peer reviews follow certain writing norms

and styles, and the content is scientific in nature, which makes supervised training on domain-specific examples a promising candidate. This raises the question: *Does the presence of peer-review-specific signals help overcome the limitations of fine-grained AI detection in peer reviews?*

**Main findings.** We summarize our key findings below.

- Pangram and GPTZero successfully identify $98.3\%$ and $95.8\%$ of fully AI-generated reviews (AI-BP and AI-EP), respectively. For fully human-written reviews, Pangram flags none of them as "AI" or "Mixed", while GPTZero flags $1\%$ of such reviews as "AI."[1]

- All detectors struggle with the Human-written AI-polished (H-AI) reviews. These reviews are compliant under a Polishing-only policy. However, Pangram and GPTZero have a non-trivially high false positive rate, classifying $3.1\%$ and $3.4\%$ of such reviews as "AI" (not even "Mixed"), respectively.

- Zero-shot open-source detectors perform notably worse compared to the proprietary ones. The best-performing of them in our experiments, Fast-DetectGPT, can identify only $70\%$ of fully AI-generated (AI-BP, AI-EP) reviews with false positive rates of $4.6\%$ and $0.2\%$ on AI-polished and human written reviews respectively. With the paper manuscript under review as additional context, their ability to detect AI-BP, AI-EP, AI-HI reviews improves, but at the cost of even higher false positives rates.

- Under humanization, Pangram and GPTZero continue to flag the vast majority of humanized AI-BP reviews as either "AI" or "Mixed" rather than "Human" ($92.8\%$ and $86.7\%$, respectively), and so remain useful in determining whether some form of AI-assistance was involved. However, surprisingly, H-AI reviews tend to be flagged as "AI" more frequently if they are humanized.

- To leverage peer-review specific signals, we measure similarity of a candidate review with several AI-generated reference reviews (given we have access to the paper manuscript). We find that these similarity scores yield different distributions across varying levels of AI assistance, at an aggregate level. However, these distributions overlap substantially and many individual AI-polished reviews receive similarity scores indistinguishable from those of fully AI-generated reviews. Therefore, this approach can not be used for reliable review-level detection.

- The narrow writing style and scientific content of peer reviews, while a natural motivation for supervised training, does not lead to much added reliability in detection. Supervised classifiers based on stylometric features and dense embeddings fail to generalize beyond the LLMs

---

[1] Considering potential data contamination concerns (detectors having seen the same human reviews in their training), these are optimistic estimates. See more in the note on contamination (§ 4.1)

whose reviews they were trained on.

- Finally, a practical note for reviewers using LLMs to polish their drafts. Uploading the submitted paper, or omitting explicit content-preservation instructions in the polishing prompt, can lead LLMs to introduce new content in rather than just polishing it. Reviews polished using these prompts are classified as "AI" significantly more often. Reviewers should therefore phrase their prompts carefully and verify the 'polished' output before submission.

Overall, we find that neither off-the-shelf detectors nor strategies that leverage peer-review-specific advantages lead to systems that can consistently classify LLM-polished reviews into the same category as human-written or mixed ones. This corroborates recent findings on the challenges of detecting mixed-authorship text in non peer-review data (Saha & Feizi, 2025; Zhang et al., 2024). This finding has a direct policy implication: *It is infeasible to enforce any policy that permits LLM usage selectively for improving flow, clarity or grammar correction.* In contrast, under a blanket No-LLM-use policy, AI-polished reviews could be misclassified as human-written, in which case these (supposedly minor) violations would go undetected. Beyond enforcement of policies, our findings carry implications on how detection performance should be interpreted and communicated. For instance, *a recent estimate that 21% of peer reviews in ICLR 2026 was fully AI-generated (Pangram Labs, 2025b) may not be accurate*, as these estimates likely include reviews written with only partial AI involvement.

**Conflict of Interest Disclosure.** A subset of Pangram API requests were supported through Pangram's Academic plan. Findings, conclusions and recommendations expressed here are those of the authors alone and do not necessarily reflect the views of our sponsors.

## 2. Related Work

Recent OpenAI and Anthropic reports on ChatGPT (Chatterji et al., 2025) and Claude (Handa et al., 2025) usage patterns indicate that people frequently delegate tasks that require analyzing, summarizing and reviewing information to AI. Writing is a dominant use case across both platforms. Several recent studies attempt to estimate the extent of AI use in writing peer reviews. Liang et al., 2024 estimate that approximately 6.5-16.9% of all the peer review text from ICLR 2024, NeurIPS 2023, CoRL 2023 and EMNLP 2023 are AI-modified beyond simple writing updates. Detecting individual AI-generated reviews remains challenging. Recently, ICML 2026 (Agarwal et al., 2026) adopted the strategy proposed by Rao et al. (2025) whereby it injected prompts in white text in manuscript PDFs instructing the AI models to embed a covert watermark in the generated review which can later be detected with strong statistical guarantees on family wise error rates.

Current post-hoc AI text detectors broadly fall into two categories: (a) supervised classifiers such as OpenAI's AI Text Classifier (OpenAI, 2023) and RADAR (Hu et al., 2023), that train neural networks on human-AI parallel datasets, and (b) training-free zero-shot methods using statistical signatures of AI-generated text such as perplexity (Solaiman et al., 2019), perplexity curvature (DetectGPT, Mitchell et al., 2023), and cross-perplexity (Binoculars, Hans et al., 2024). However, these detectors often fail to identify AI-generated reviews, with evaluations showing that up to 40% of purely AI-generated reviews are misclassified as human-written (Yu et al., 2025). Our study finds that more recent proprietary AI detectors like Pangram (Emi & Spero, 2024; Thai et al., 2025) and GPTZero (Tian & Cui, 2026), trained on very large scale datasets with human content and their AI-generated mirrors, have substantially improved detection of fully AI-generated content, a finding corroborated by independent benchmarks (Jabarian & Imas, 2025). Several studies estimating the prevalence of AI involvement in peer reviews and academic writing rely on the outputs of these detectors (Russo et al., 2025; Elazar & Antoniak, 2026). Meanwhile, ChatGPT usage data (Chatterji et al., 2025) indicates that nearly two-thirds of writing-related interactions in the platform involve modifying user-provided texts rather than generating new text from scratch. This suggests that AI involvement in peer reviews is better characterized as a spectrum, ranging from light stylistic polishing to completely generating reviews, which we model in our evaluation.

Prior work has documented misclassification of AI-polished text (Saha & Feizi, 2025; Zhang et al., 2024) as entirely AI-generated, but peer review presents high stakes policy implications as well as structural advantages to the problem of AI text detection. Our work evaluates promising directions for fine-grained AI detection that have emerged recently: for instance, Thai et al. (2025) introduce Editlens (used in Pangram 3.0) which is a recipe to predict the extent of AI edits applied to a source human document that produces a given text. Additionally, our work also explores whether unique opportunities in the peer review setup (e.g. domain specificity, availability of the paper under review) can help circumvent concerns of false positives on AI-polished human content and discusses their implications on the enforceability of Polishing-only policies in peer reviews.

Finally, we also clarify that our position is not against the use of LLMs in reviewing. In fact, LLMs can be particularly useful in ensuring scientific rigor (Liu & Shah, 2023; Xi et al., 2025; Shah, 2025b, §10.1.2 and §11.3.2) although LLM reviewers need to be evaluated appropriately (Baumann et al., 2026; Shah, 2025a, Part 1). Our work simply aims to evaluate current policies adopted by journals and conferences, and alongside, provide more evidence to supplement broader discourse on this topic.

*Table 1.* Distribution of papers and human-written reviews.

| Conference | Papers | Human Reviews |
|------------|--------|---------------|
| CONLL 2016 | 22 | 39 |
| ACL 2017 | 137 | 275 |
| ICLR 2017 | 427 | 1303 |
| NeurIPS 2013-17 | 500 | 1882 |
| **Total** | **1086** | **3499** |

## 3. Evaluation Framework

In this section, we outline the creation of the peer-review dataset and the metrics used to evaluate various detectors.

**Dataset of human-written reviews.** Our study requires peer reviews authored without AI assistance. To be certain that reviews are human written, we use pre-2020 review data (before GPT-3's release). The PeerRead dataset (Kang et al., 2018) comprises $14,000$ paper drafts and $10,000+$ expert reviews from conferences including NeurIPS, ACL, ICLR, and CoNLL, all from 2017 or earlier. We work with a subset of 1086 papers (500 papers from NeurIPS and full set from ACL, ICLR, CoNLL) and 3499 corresponding human-written reviews for papers from these venues (Table 1).

**Reviews generated with different levels of Human involvement.** We construct a dataset of reviews that systematically vary in their degree of human involvement. Specifically, we generate reviews at five levels of human involvement, ranging from completely AI-generated reviews to fully human-authored reviews. While the exact forms of AI assistance adopted by reviewers in practice remain undetermined, these levels are constructed to simulate plausible scenarios. In the descriptions that follow, we motivate each level and specify the inputs provided to the LLM.

- **AI-generated with Basic Prompts (AI-BP):** This level captures the most disengaged reviewer. To generate AI-BP reviews, we prompt an LLM with just the paper and the conference's official reviewing guidelines.

- **AI-generated with Elaborate Prompts (AI-EP):** This level simulates a lazy reviewer who is just resourceful enough to grab general reviewing advice and best practices from the internet and incorporate them into the prompt, but does not contribute any original opinion or content of their own. We aggregate best practices from five conference-issued reviewer best-practice documents (e.g., ACL 2017 last minute reviewing advice; full list in Appendix H). The prompt includes these best practices along with the paper and reviewer guidelines.

- **AI-generated with Human Input (AI-HI):** This level represents the case where reviewers identify key assessment points after skimming the paper and instruct an LLM to elaborate on them. We simulate this via a two-step pro-

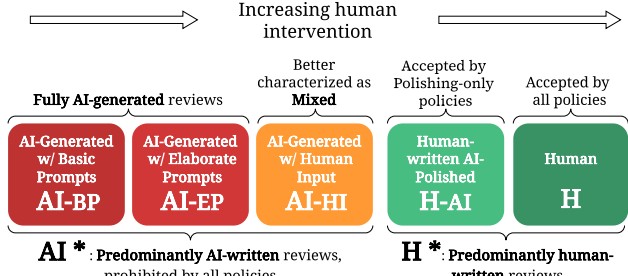

*Figure 1.* Levels of AI-assistance.

cess: first, an LLM extracts key assessment points from a human-written review; these key points are then included in the prompt for an LLM to elaborate upon. The prompt also includes the paper text and reviewer guidelines.

- **Human-written AI-polished (H-AI):** This is the only acceptable form of LLM use under "Polishing-only" policies. Here, the LLM is given just a human-written review, without the paper manuscript or reviewing guidelines, and is instructed to improve grammar, flow and clarity while preserving structure, meaning and technical content. In addition to explicitly including content-preservation instructions in the prompt, we filter out reviews whose length exceeds $1.25$ times that of the original human review to limit the LLM's involvement to polishing only.[2]

- **Completely Human (H):** This level comprises original human-authored peer reviews from the PeerRead dataset.

An overview of the above levels is available in Figure 1. In the rest of the paper, we use the term **AI**\* to refer to the first three levels combined (AI-BP, AI-EP and AI-HI). AI-BP and AI-EP are clear violations of Polishing-only policy, while H-AI is permitted. One might argue that for AI-HI reviews, the LLM's involvement is limited to stitching together the human-written key assessment points into prose and hence can be considered as AI-polished. However, unlike H-AI reviews, the AI-HI prompt explicitly instructs the LLM to elaborate on these key points, going beyond merely polishing the reviews. The LLM also has access to the paper manuscript to expand the key points. On average, the length of the AI-HI reviews (679 words) is roughly 4 times the length of keypoints (163 words) used to generate them. Though these reviews are best classified as "mixed", while reporting performance of detectors, we consider them under the "positive" class (violation under Polishing-only policy). For detectors whose predictions include a "mixed" label, we separately report the full distribution of predicted labels. To summarize, while reporting performance of de-

---

[2] All H-AI reviews used for the results in the main paper are generated using this procedure. We acknowledge that real-world polishing behaviour can be more diverse, and report some preliminary results on additional plausible variants in Appendix C.

tectors concisely in terms of true and false positive rates, AI* reviews are considered "positives" and the rest, together denoted as **H***, are considered "negatives."

For each paper, we generate reviews using GPT-4o and Llama-3.3-70B-Instruct models, with one prompt per level. This results in 18, 122 AI-generated reviews, which we refer to as the **easy subset**. The easy subset provides higher coverage in terms of number of papers, but limited diversity in the choice of models generating the reviews. Therefore, additionally, for a subset of 158 papers (18 from CoNLL 2016 and 20 from each of the remaining conferences), we generate reviews using GPT-5, Gemini-2.5-pro, Gemma-3-27b-it, Qwen-3-30B-thinking and Llama-3.1-70B-instruct. For this subset, we employ at least four distinct prompts per level, resulting in 26, 535 AI-generated reviews, which we refer to as the **hard subset** (see Appendix H for full set of prompts). With newer models and diverse prompts, the hard subset more closely captures AI assistance in the wild and yields a more challenging evaluation. Combined with 3499 human-written reviews, our dataset contains 48, 156 including both the easy and hard subsets.

**Humanized Reviews.** "Humanization" refers to the paraphrasing of AI-generated text, with the deliberate aim of mimicking human writing to evade detection (Shi et al., 2023; Wang et al., 2024; Masrour et al., 2025). This is different from the collaborative paraphrasing scenario that H-AI represents. Numerous automated tools are publicly available and commonly used for this purpose. We employ Undetectable AI,[3] one of the most widely-used commercial AI humanizers with an API, to humanize reviews from two key levels: AI-BP and H-AI. Humanizing AI-BP reviews simulates the scenario of a lazy reviewer trying to present a fully AI-generated review as their own. We also consider a second plausible scenario, where an overly cautious reviewer, who has polished their draft with AI (H-AI), chooses to pass it through a humanization tool to minimize the risk of detection. Including 2000 such humanized reviews, the size of our dataset becomes 50, 156.

**Metrics and Calibration.** To evaluate a detector, we report the fraction of reviews predicted as AI-generated ("positive" class) at each level. This corresponds to true positive rate (TPR) for AI* reviews and false positive rate (FPR) for H* reviews following our earlier discussions outlining the rationale of deciding which levels constitute violation of Polishing-only policy. For open-source zero-shot detectors that output scalar detection scores, we use a calibration set of human-written reviews from NeurIPS 2013-2015 to adjust the thresholds for the "positive" class. The thresholds are chosen to ensure 0% FPR on the calibration set. We evaluate these detectors on reviews of conferences after 2016. The rationale is to simulate a situation where ground truth

from past conferences is used to calibrate the detectors. For fair comparison with other detectors, we also include examples from NeurIPS 2013-2015 in the evaluation set, ensuring none of them belong to the same paper as any review in the calibration set (Table 2). For Pangram and GPTZero, which produce three-class predictions ("AI", "Mixed", "Human"), we report TPR and FPR in Table 2 by treating only reviews labeled "AI" as the positive class, collapsing "Mixed" with "Human". This conservative choice minimizes false positives on policy-compliant reviews. Additionally, we also examine the full distribution of predicted labels across all three classes, reported as confusion matrices in Figure 2, to provide a more complete picture of detector performance.

## 4. Experiments & Results

In this section we discuss the experiments we conducted and enumerate our findings.

### 4.1. Can off-the-shelf AI detectors distinguish between different levels of AI assistance in peer reviews?

We evaluate a set of 3 open-source and 2 proprietary detectors on our review dataset, selected to represent the landscape of currently available AI detection tools (hyperparameter settings and other details in Appendix D.1).

We find that **Pangram correctly flags more than** 90% **of AI-generated reviews with minimal to no human input (AI*** **reviews), while maintaining** 0% **FPR on human reviews.** Pangram and GPTZero both return fine-grained labels of AI involvement, "AI", "Mixed" or "Human". Figure 2 presents the distribution of reviews in each level across these labels on the hard subset (for similar plots for the easy subset see Appendix D.2). GPTZero also classifies high fraction of AI* reviews as either "AI" or "Mixed", but unlike Pangram, has non-zero FPR on human-written reviews. In contrast, **open-source zero-shot AI detectors cannot detect AI*** **reviews generated with newer models and varied prompts.** Fast-DetectGPT, the best performing open-source detector in our experiments, has near perfect detection rates, on par with commercial detectors on the easy subset. However, in the hard subset, with more modern LLMs and varied prompts, its ability to detect AI* reviews drops sharply, limiting its practical utility for policy enforcement.

Further, **Pangram and GPTZero classify a non-trivial fraction of human-written but AI-polished (H-AI) reviews as AI-generated.** One strategy to use the predicted labels from Pangram and GPTZero ("AI," "Mixed," or "Human") for enforcing Polishing-only policies is to penalize only the reviews which are classified as "AI". In Table 2, we treat "Mixed" in the same category as "Human" to report TPRs and FPRs of the detectors, ensuring they are optimized for low FPR on policy-compliant reviews. Even under this

---

[3] https://undetectable.ai/

*Table 2.* Performance of detectors with and without paper context. Pangram and GPTZero are context-agnostic by design. We note that in many cases, zero-shot detectors with paper context yield marginal improvements over the no-context case (* indicates statistically significant difference, more details in Appendix E.1). Pangram and GPTZero perform significantly better. However, both of them classify a **non-trivially high percentage** of human-written but AI-polished (H-AI) reviews as AI-generated, even when "Mixed" labels are regarded as human-written (more in Section 4.1 and Fig. 2). Hence, Polishing-only policies are currently not enforceable.

| Method | Easy subset | | | | Hard subset | | | | Human |
|---|---|---|---|---|---|---|---|---|---|
| | TPR (%)↑ | | | FPR (%)↓ | TPR (%)↑ | | | FPR (%)↓ | FPR (%)↓ |
| | AI-BP | AI-EP | AI-HI | H-AI | AI-BP | AI-EP | AI-HI | H-AI | H |
| Pangram 3.0 | 100.0 | 100.0 | 100.0 | 3.0 | 97.0 | 99.3 | 92.6 | **3.1** | 0.0 |
| GPTZero | 96.7 | 93.3 | 91.0 | 3.0 | 96.0 | 95.8 | 89.4 | **3.4** | 1.0 |
| LogLikelihood | 97.8 | 96.4 | 72.4 | 0.4 | 46.0 | 40.9 | 30.5 | 0.0 | 0.0 |
| with paper context | 98.9 | 99.6 | 91.6* | 1.6 | 59.8* | 50.5* | 43.7* | 4.2* | 0.4 |
| Fast-DetectGPT | 100.0 | 100.0 | 97.5 | 3.5 | 72.1 | 68.2 | 63.1 | 4.6 | 0.2 |
| with paper context | 100.0 | 100.0 | 99.3* | 9.0* | 75.3* | 73.9* | 68.2* | 9.5* | 0.9 |
| Binoculars | 51.1 | 50.7 | 39.6* | 0.0 | 21.6 | 22.3 | 14.1 | 0.1 | 0.0 |
| with paper context | 50.7 | 53.3 | 33.7 | 0.0 | 20.5 | 20.0 | 14.3 | 0.2 | 0.0 |

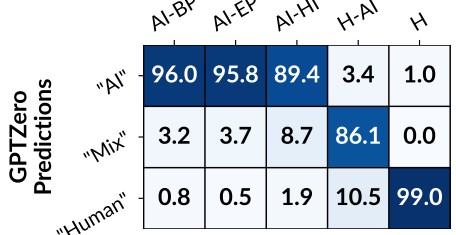

*Figure 2.* Confusion matrices denoting % of reviews classified as "AI", "Mixed", "Human" by Pangram & GPTZero on hard subset.

favorable setup, more than 3% of H-AI reviews are penalized in hard subset (see Figure 2).[4] On the other hand, under a No-LLM-use policy, reviews classified as "Mixed" should also be penalized. Because H-AI reviews are prohibited under this policy, enforcing it with Pangram (GPTZero) would mean that 74.7% (10.5%) of H-AI reviews will go undetected, since they are classified as "Human."

If conference organizers were to enforce Polishing-only policies with these detectors, an unacceptably large fraction of policy-compliant reviews would be flagged and penalized. Therefore, **"Polishing-only" policies are not currently enforceable**. We note that **these conclusions are robust to potential data contamination**. It is conceivable that the training data of the LLMs used to generate reviews or the AI text detectors may have included the original human reviews from PeerRead dataset. Firstly, this may enable LLMs to produce more human-like reviews making our evaluation dataset more challenging. Since some detectors (the proprietary ones) are able to detect purely AI-generated reviews regardless, this scenario does not weaken our conclusions. Secondly, if detectors like Pangram and GPTZero saw these reviews during their training on web-scale human-AI parallel data, our results constitute an optimistic estimate of their

performance in the wild, despite which they fail to meet the accuracy standards required for enforcing Polishing-only policies. Therefore, potential contamination does not affect our conclusions. A more subtle concern is that possible contamination could cause our evaluation to underestimate Pangram's false positive rate on fully human-written reviews. However, Pangram's publicly available technical documentation independently reports a false positive rate of approximately 1 in 10,000 (Emi, 2025) on fully human-written content. If this self-reported FPR is to be trusted, then tjh 0% FPR we observe on human reviews is unlikely to be an artifact of contamination.

Surprisingly, we observe that **LLMs occasionally introduce new content to the human review when instructed to polish it for improving grammar and clarity** (more details in Appendix A). We find that three factors pertaining to the structure of the polishing-only prompt lead to such behaviour: (1) attaching the paper manuscript, (2) omitting an *explicit instruction* to preserve technical content, and (3) specifying a generous word limit. These findings offer actionable insights for reviewers seeking to use AI for polishing while avoiding false accusations, **reviewers should phrase their prompts clearly, with explicit instructions to not include new content, and should verify the re-**

---

[4]Model wise breakdown of FPR in Appendix B

**sulting outputs carefully**. For our own pipeline, we adopt safeguards (see Appendix A.1) to mitigate the LLM's contribution beyond polishing. For the $\sim 3\%$ of our H-AI reviews flagged as fully AI-generated by commercial detectors (Table 2), we manually verify that the misclassifications are not due to new technical content and are genuine false positives.

LLM-polished reviews generated without careful application of the aforementioned safeguards are flagged as fully AI-generated for $59.7\%$ of cases by Pangram and $52.4\%$ by GPTZero. They are part of a broader spectrum of reviews apart from H-AI which, while not strictly polishing-only, are more accurately characterized as mixed human-AI authorship rather than fully AI generated. This also includes AI-HI reviews which are classified as fully AI-generated more than $90\%$ of times by Pangram. This tendency to label reviews with substantial human-authored input as "AI" rather than "Mixed" has important implications for interpreting detector-based estimates of AI-generated reviews. Pangram Labs recently used their detector to analyze ICLR 2026 reviews and submissions, claiming that $21\%$ of reviews were fully AI-generated (Pangram Labs, 2025b) and flagging many individual reviews and submitted papers as "fully-AI generated" (Pangram Labs, 2025a). However, our findings suggest that **both these claims likely misrepresent the extent of fully AI-generated content**: the former may include a substantial fraction of mixed-authorship reviews, and a non-trivial number of the latter might be misclassifications given the scale of ICLR.

## 4.2. What is the impact of "humanization" (adversarial paraphrasing)?

In our earlier experiments, the detectors already struggle with the hard subset, even without humanization. We therefore focus the analysis on the easy subset, where baseline performance is sufficiently strong to reveal the impact of humanization. These results are summarized in Table 3.

We find that **detectors are impaired by humanization of both fully AI-generated and AI-polished reviews.** Pangram and GPTZero see a significant drop in their ability to detect fully AI-generated reviews after humanization. More surprisingly, humanization results in higher false positives for human written but AI polished reviews relative to the pre-humanization setting, marginally for Pangram but dramatically for GPTZero. FastDetect, Binoculars and LogLikelihood, on the contrary, classify all reviews as human-written post humanization (both AI and H-AI) (Table 3).

**Both Pangram and GPTZero can deal with humanization under No-LLM-use policy.** Pangram classifies $44.6\%$ of humanized AI-BP reviews as "AI", and an additional $48.2\%$ as "Mixed". Therefore, under a No-LLM-use policy, where "Mixed" reviews are also unacceptable, Pangram successfully flags $92.8\%$ of humanized policy violations.

*Table 3.* Post humanization, open-source detectors classify all reviews as human. Similarly, Pangram and GPTZero suffer notable drops in their ability to detect AI-generated reviews if they are humanized. Importantly, H-AI reviews are more frequently detected as AI by Pangram and GPTZero after humanization.

| Detector | AI-BP TPR (%) ↑ | | H-AI FPR (%) ↓ | |
| --- | --- | --- | --- | --- |
| | **pre** humanization | **post** humanization | **pre** humanization | **post** humanization |
| Pangram | 100 | 44.6 | 3.0 | 3.63 |
| GPTZero | 96.7 | 86.7 | 3.0 | 65.6 |
| LogLikelihood | 96.7 | 0.0 | 0.0 | 0.0 |
| FastDetect | 100 | 0.0 | 2.6 | 0.0 |
| Binocular | 51.1 | 0.0 | 0.0 | 0.0 |

GPTZero, on the other hand, classifies $86.7\%$ of humanized AI-BP reviews as "AI" but none as "Mixed". Hence, unlike Pangram, a blanket ban on LLM-assistance will not help its ability to detect violations. However, its humanized H-AI FPR of $65.6\%$ renders it unsuitable for enforcing a Polishing-only policy. This makes No-LLM-use policy more enforceable in presence of humanization tools.

## 4.3. What additional context does the peer review setting provide, and is that useful for AI detection?

We identify two advantages of the peer review setting that can be potentially leveraged to improve detection performance: (1) availability of the manuscript of the paper under review, and (2) restriction to a specific writing style and the relatively narrow domain of scientific peer reviews. We begin with the former by examining two strategies to make use of the paper manuscript.

**(a) Conditioning likelihood on the paper contents.** The paper PDF is often the majority of the prompt that goes into an LLM to generate a review. Zero-shot detectors rely on probability of the candidate text under a scoring language model. LogLikelihood, Fast-DetectGPT and Binoculars fall under this class of detectors. As a simple modification, we condition the scoring model's probabilities on the paper's context, formed by concatenating the abstract, introduction, and conclusion. The results are summarized in Table 2 alongside their regular context-agnostic variant. Incorporating paper context yields statistically significant improvements in detection rates of AI* for LogLikelihood and Fast-DetectGPT. However, even with these gains, TPR remains below $80\%$ on the hard subset, with significantly higher false positives on H-AI reviews. Therefore, we conclude that **the additional context of paper manuscript yields measurable gains in the performance of zero-shot detectors, albeit not to an extent that makes them usable.**

**(b) Similarity with AI-generated reference reviews.** In this approach, we first generate several reviews by prompting LLMs with the paper. Higher similarity with these AI

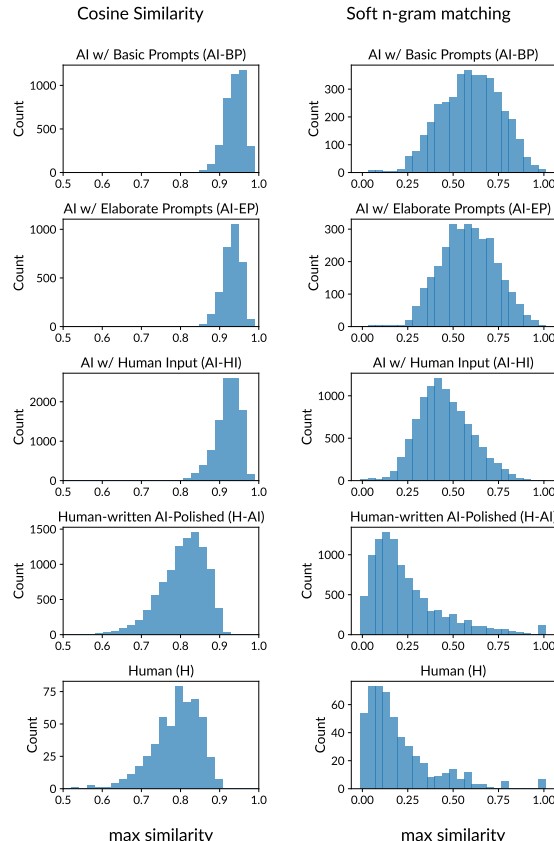

*Figure 3.* Distribution of similarity (maximum over all AI-generated references) across levels of AI assistance. Many individual AI-polished reviews receive similarity scores indistinguishable from those of AI* reviews.

*Table 4.* Classifier trained on similarity to AI generated references as features. High error rates across-the-board indicate infeasibility of individual review level predictions with these features.

| | TPR (%) ↑ | | | FPR (%) ↓ | |
|---|---|---|---|---|---|
| **Similarity metric** | AI-BP | AI-EP | AI-HI | H-AI | H |
| Cosine similarity | 98.5 | 98.2 | 90.8 | **8.2** | **4.0** |
| Soft n-gram match | 98.7 | 98.5 | 93.8 | **4.0** | **2.0** |
| Soft keypoint match | 98.8 | 97.7 | 88.9 | **4.4** | **3.0** |

*Table 5.* Performance of supervised classifiers on examples generated with models and prompts seen during training. In this setting, full finetuning of a RoBERTa classifier yields better performance than off-the-shelf detectors, including proprietary ones.

| | TPR (%) ↑ | | | FPR (%) ↓ | |
|---|---|---|---|---|---|
| **Classifier** | AI-BP | AI-EP | AI-HI | H-AI | H |
| Subset: **Easy** | | | | | |
| Stylometric | 99.3 | 99.6 | 98.6 | 1.3 | 1.1 |
| RoBERTa-base | 100.0 | 100.0 | 100.0 | 0.9 | 0.0 |
| Subset: **Hard** | | | | | |
| Stylometric | 99.1 | 97.3 | 95.6 | 4.3 | 0.0 |
| RoBERTa-base | 100.0 | 100.0 | 99.9 | 1.4 | 0.0 |

references is an indicator of greater AI involvement ([Yu et al., 2025](#)). Related work has also explored similarity-based measures for assessing AI involvement in text. EditLens ([Thai et al., 2025](#)) quantifies AI editing by estimating similarity to a hypothetical human-written version of the same text. Our approach is conceptually similar, but avoids reliance on such a hypothetical human reference and instead leverages multiple AI-generated references.

**Similarity metrics:** We start with the cosine similarity between the pretrained embeddings of the candidate and the reference review ([Yu et al., 2025](#)). Additionally, towards a more granular measure, we decompose the candidate and reference review into smaller semantic units and measure what fraction of candidate units have a close match in the reference. The metric is called **Soft n-gram matching** if the semantic units are n-grams ([Thai et al., 2025](#)), and **Soft keypoint matching** if the semantic units are key ideas extracted from the text (implementation details in Appendix [E.2](#)).

For every paper, we use 3 models (GPT-5, Gemma-3-27b-it, Qwen-3-30B-thinking), 3 prompts and 5 roll outs per model-prompt pair to generate 45 references. We find that

**predictions about individual reviews, based on similarity to AI-generated references, are unreliable.** We observe a consistent monotonic shift in the distribution of similarities towards lower values as we move from completely AI-generated (AI-BP) to completely human-written (H) reviews (Figure [3](#)). However, these distributions overlap substantially enough that individual reviews cannot be reliably classified. To further illustrate this limitation, we train a five-class XGBoost ([Chen & Guestrin, 2016](#)) classifier using similarity scores with all 45 references as the feature vector, sorted to ensure permutation invariance across reference orderings. At inference, examples predicted as AI-BP, AI-EP or AI-HI are mapped to the "positive" (AI-generated) class.[5] With this, more than 4% of AI-polished reviews are misclassified as AI-generated (Table [4](#)) with each of the similarity metrics. Additionally, and concerningly, we see non-zero FPR on fully human-written (H) reviews. This confirms the unreliability of similarity-based detection. We speculate that pretrained embeddings may not adequately capture the differences in reasoning, style and argumentation that distinguish human writing from AI.

The above limitations of strategies leveraging the paper manuscript motivates us to turn to the second peer-review-specific advantage: the restricted domain of scientific reviews. Supervised classifiers for AI text detection are known to excel on in-domain data but struggle with text drawn from a wider range of domains ([Bakhtin et al., 2019](#); [Uchendu](#)

---

[5]The definitions of TPR and FPR are adjusted accordingly

*Table 6.* Performance of supervised classifiers on reviews generated by a model held out during training (a realistic out of distribution setting). Here, supervised classifiers can exhibit exhibit inconsistent generalization, either failing to detect AI* reviews or produce non-trivially high false positive rates on H-AI reviews. **Dark red** represents *some* selected concerning values.

| Holdout model | Stylometric | | | | | RoBERTa-base | | | | |
|---|---|---|---|---|---|---|---|---|---|---|
| | TPR (%) ↑ | | | FPR (%) ↓ | | TPR (%) ↑ | | | FPR (%) ↓ | |
| | AI-BP | AI-EP | AI-HI | H-AI | H | AI-BP | AI-EP | AI-HI | H-AI | H |
| GPT-5 | 98.7 | 97.5 | 93.0 | **3.6** | 0.0 | 100.0 | 100.0 | 100.0 | 1.6 | 0.0 |
| Gemini-2.5-pro | 90.7 | **85.8** | 76.2 | **3.7** | **1.0** | 100.0 | 100.0 | 96.3 | **4.6** | 0.0 |
| Gemma-3-27b-it | 99.3 | 95.8 | 93.8 | **3.5** | **1.0** | 100.0 | 100.0 | 95.5 | 0.0 | 0.0 |
| Llama-3.1-70B-instr | **45.3** | 39.2 | 52.2 | 1.3 | 0.0 | 92.0 | **87.5** | 86.3 | 0.3 | 0.0 |
| Qwen-3-30B-thinking | **88.0** | 85.0 | 81.8 | **11.0** | 0.0 | 100.0 | 100.0 | 100.0 | **11.5** | 0.0 |

et al., 2020; Pu et al., 2023). However, peer reviews, and academic discourse in general, follow a certain writing style and constitute a relatively narrow domain. Supervised training is a strong candidate to leverage these stylistic and domain regularities. We explore two distinct sets of features.

**(a) Stylometric features.** Stylometric and linguistic features have been previously used for authorship attribution and AI-text detection tasks (Opara, 2025; Reviriego et al., 2024; Agrahari et al., 2025). We choose a set of 38 such features (exhaustively listed in Appendix E.3) capturing lexical diversity (e.g. type-token ratio, n-gram uniqueness, etc.), POS tags (e.g. verb, abstract noun percentage, etc.), readability (e.g. Flesch reading ease etc.) and other statistics.

**(b) Transformer-based representations.** Supervised classifiers built on pretrained transformer representations have been found to be effective for AI-generated text detection, particularly in narrow domains (Zellers et al., 2019; Rodriguez et al., 2022; Marchitan et al., 2024). We do a full finetuning of a RoBERTa model (Liu et al., 2019) with a 5-label classification head with overlapping segments of fixed length from review texts. Final prediction is obtained through majority voting over segment-level predictions.

We first report performance in the in-distribution setting where training and test sets are disjoint subsets of the same split (easy or hard) in Table 5. We conduct further experiments to simulate a more realistic scenario where the classifier encounters reviews generated by unfamiliar models during deployment. Table 6 presents these results, where each row corresponds to holding out one model entirely from the training data while training on reviews from all other models. For both the classifiers described above, following earlier discussion, we treat reviews classified as AI-BP, AI-EP and AI-HI as "positive" or AI-generated.[5]

**In the in-distribution setting, supervised classifiers trained on the review dataset surpass the performance of off-the-shelf detectors**, including proprietary ones (Table 5). The stylometric classifier achieves 95.6% detection of AI-HI reviews but a 6.4% false positive rate on H-AI reviews. The RoBERTa classifier improves upon this, re-

ducing false positives on H-AI reviews to just 1.4% while detecting 99.9% of AI-HI reviews, outperforming the proprietary detectors. However, **when evaluated on reviews from models held out during training, both classifiers exhibit inconsistent generalization across held out models** (Table 6). For instance, when Llama-3.1-70B is held out, the RoBERTa classifier's ability to detect AI* reviews drops substantially (as low as 87.5% on AI-EP). For others, such as Qwen-3-30B, the classifier maintains near-perfect detection of AI* reviews, but at the cost of an 11.5% false positive rate on H-AI reviews. While the classifier generalizes well to certain held-out models (e.g., GPT-5 and Gemma-3), conference organizers cannot know in advance which models reviewers will use given the frequency of new LLM releases. The unpredictable generalization behaviour implies that to use supervised detectors for enforcing policies in the wild, **their training must keep up with the rapid pace of new model releases** since reviewers can routinely use systems unseen during detector training.

## 5. Conclusions

We examined the feasibility of enforcing policies that permit LLM use for polishing peer reviews while prohibiting more substantial AI assistance. We curated a dataset of more than 50,000 LLM-generated, LLM-assisted and purely human-written paper reviews, and our analysis of popular text detectors on these reviews revealed that detectors misclassify a non-trivial fraction AI-polished reviews as AI generated. Our work has two key implications relevant to policy design:

- Policies allowing reviewers to use LLMs only for polishing cannot be reliably enforced with current AI-generated text detectors, as these are unable to consistently distinguish policy-compliant AI-polished reviews from policy-violating AI-generated ones.

- Prior estimates of AI participation in peer reviews based on outputs of AI-text detectors may conflate mixed-authorship reviews with fully AI-generated ones and potentially overstate the extent of policy violations.

## Acknowledgments

The work of NBS was supported by NSF 1942124, 2200410 and ONR N000142512346. A subset of Pangram API requests were supported through the Pangram Academic Plan. Findings, conclusions and recommendations expressed here are those of the authors alone and do not necessarily reflect the views of our sponsors.

## Impact Statement

The most direct impact of our work is to inform policy decisions at academic conferences and journals. Our findings demonstrate that polishing-only policies, while well-intentioned, are practically infeasible to enforce without wrongly penalizing honest reviewers. In late 2025, Pangram Labs publicly claimed that $21\%$ of ICLR 2026 peer reviews were fully AI-generated (Pangram Labs, 2025b), sparking widespread concern about the integrity of the peer review process. This was covered in Nature (Naddaf, 2025) and discussed extensively. Our findings show that such claims must be interpreted with caution, as they may substantially overestimate violations. We call for better transparency about false positive rates when detector outputs are publicized.

Our evaluation is a step towards better transparency about false positive rates, and indirectly impacts other areas that see use of AI text detectors, for instance, detecting AI use in student assignments. We also acknowledge that greater transparency comes with a potential downside. Public awareness of high false positive rates may enable non-compliant reviewers to claim innocence citing detector unreliability, making it harder to identify actual violations and worsening the enforcement of polishing-only policies.

Finally, we also explored several strategies to enhance existing open-source detectors by leveraging advantages unique to the peer-review setting, such as manuscript context, domain constraints, and similarity to AI-generated reference reviews. However, none of these approaches yielded broadly reliable fine-grained detection. That said, these signals did yield measurable improvements in certain settings, and future work could improve upon them. Such improvements may have implications beyond peer review to other scenarios where additional contextual information about the generation process is available.

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

# Appendices

## Contents

## A. LLMs Can Introduce New Technical Content in Response to Polishing-Only Prompt

Surprisingly, we observe that **LLMs occasionally introduce new content to the human review when instructed to polish it for improving grammar and clarity**. We observe that three factors pertaining to the structure of the

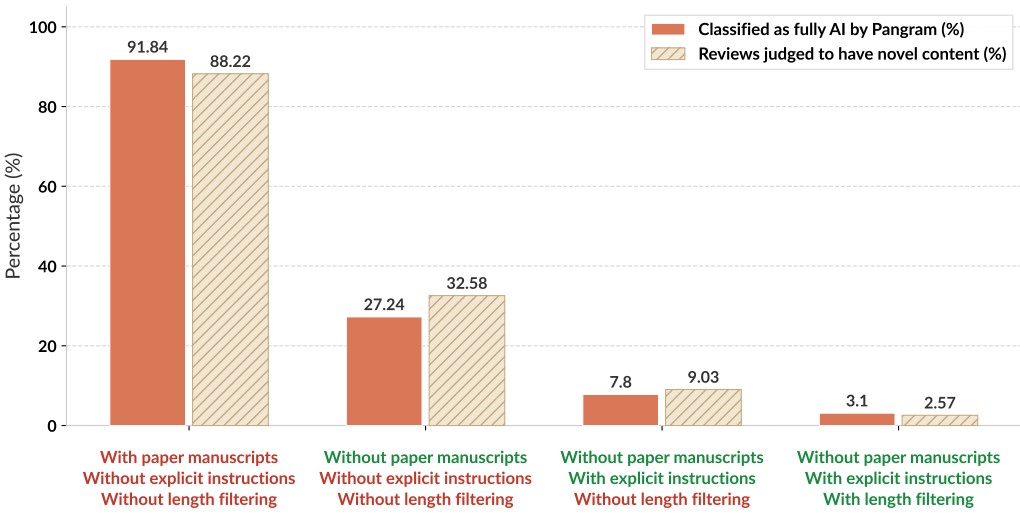

*Figure 4.* **LLMs inadvertently introduce new content rather than merely "polishing":** Including paper manuscript, omitting *explicit content-preservation instruction* and specifying generous word limits in the prompt leads to LLMs introducing new content in the "polished" review. Interestingly, such reviews are more likely to be flagged as AI-generated by Pangram.

polishing-only prompt lead to such behaviour: (1) attaching the paper manuscript, (2) omitting *explicit instruction* to preserve technical content, and (3) specifying a word limit that is higher than the length of the human-written draft review. In the third case, LLMs tend to treat the word limit as the target length as opposed to an upper bound and generate longer polished reviews.

We examine these effects systematically at scale by employing Gemini-2.5-pro as a judge to flag if a polished review introduces new content relative to the original human review. We pair source human reviews (Review A) with their LLM-polished counterparts (Review B) and ask judge: *Does Review B introduce any new technical information that is not present in Review A?* Since summaries may be expanded for comprehensiveness rather than substance, we ignore any new information introduced there and consider only sections requiring substantive human judgement. Additional reliability checks of judge responses are outlined in A.2.

We study four polishing strategies where the generation is subject to increasingly stricter safeguards—starting with prompts featuring all the three above-mentioned properties and progressively removing one at a time. For these strategies, Figure 4 reports the percentage of polished reviews judged to include new content, alongside the fraction of reviews classified as "AI" by Pangram. Clearly, both these percentages are highly correlated.

### A.1. Mitigation in our data generation

Above observations motivate our choice of prompts (explicit content preservation instructions, no access to paper manuscript) and post-processing (filtering out reviews $1.25$ times the original length) for constructing our H-AI reviews to limit the LLM's role to polishing only. The prompts are listed in Appendix H. It is important to note that all H-AI results reported elsewhere in the paper use reviews generated following these safeguards.

### A.2. Validating judge responses

To assess whether Gemini can serve as a reliable judge, we perform two complementary validation checks.

**Inter-annotator agreement.** We randomly sample 50 review pairs, 25 labeled "yes" and 25 labeled "no" by the LLM judge, and present them to two authors of this paper as independent human annotators. We compute Cohen's $\kappa$ (Cohen, 1960) to measure inter-annotator agreement between the two human annotators, as well as the agreement between the Gemini judge and each human annotator (Table 7). For both human annotators, LLM-human agreements are "substantial" (Table 8, Landis & Koch, 1977), which is consistent with the human-human agreement.

**Gold standard annotations.** We construct a gold-standard annotation set for the same 50 pairs using the human annotations. When both annotators concur, we adopt that label. In cases of disagreement, the two annotators assign a consensus label through discussion. We subsequently compare the predictions of Gemini against this gold standard and find

*Table 7.* Inter-annotator agreement and Gemini-as-a-judge accuracy on whether LLM-polished reviews introduce novel content relative to their source human reviews, with two independent human annotators.

| Cohen's $\kappa$ (95% CI) | | | Gold standard annotations |
|---|---|---|---|
| Human A - Human B | LLM - Human A | LLM - Human B | Gemini Judge Accuracy |
| 0.68 [0.47, 0.88] | 0.76 [0.57, 0.92] | 0.76 [0.60, 0.92] | 88.0 |

*Table 8.* Interpretation of Cohen's $\kappa$ (Landis & Koch, 1977)

| Cohen's $\kappa$ | Agreement |
|---|---|
| $\leq 0.00$ | Poor |
| $0.00 - 0.20$ | Slight |
| $0.21 - 0.40$ | Fair |
| $0.41 - 0.60$ | Moderate |
| $0.61 - 0.80$ | Substantial |
| $0.81 - 1.00$ | Almost Perfect |

that it achieves 88% accuracy (Table 7).

### A.3. Example of new content in "polished" review

New content additions are denoted in green. The polishing prompt used to generate this H-AI example lacked explicit instruction for content preservation. Blue highlighting indicates cases where the content is preserved but the sentence structure or ordering of ideas differs.

**Human-written review excerpt:** ... Pros: The few-shot learning results look good, but I'm not an expert in this area. The idea of using a "double" variational bound in a hierarchical generative model is well presented and seems widely applicable. Questions: When training the statistic network, are minibatches used? ... If your statistic network is trained on minibatches, it won't be able to learn this characterization, because a given minibatch will be missing some of the examples from the dataset ... Suggestions: Hierarchical forecasting (electricity/sales) could be an interesting and practical use case for this type of model.

**LLM-"Polished" review excerpt:** ... Decision: Reject. The paper presents an interesting idea, but the experimental results do not convincingly demonstrate a significant advance ... Supporting Arguments: The core idea of learning representations of datasets is promising. The paper clearly articulates the motivation and positions the work within the relevant literature ... The results on the synthetic 1D distributions and spatial MNIST ... The most significant results are presented for the Omniglot and YouTube Faces datasets ... a more detailed analysis of the performance gains, would be necessary to justify acceptance ... it's unclear whether the statistic network sees all datapoints within each dataset in a minibatch. If not, this introduces a bias in the gradient estimation. ... Additional Feedback: The paper would benefit from more detailed ablation studies ... The choice of the sample mean as the pooling method in the statistic

network seems somewhat arbitrary ... In the related work section, the connections to multiple-instance learning (MIL) could be explored in more detail ... The notation could be slightly improved for clarity. ... Questions for the Authors: 1. During training of the statistic network, does a minibatch contain all data points within each selected dataset, or is the minibatch constructed from individual data points across multiple datasets? ... Have they considered any strategies to mitigate this bias, such as using a larger batch size or incorporating a mechanism to ensure that all data points are considered over time

## B. Model-wide Breakdown of H-AI FPR

Pangram and GPTZero misclassify 3.1% and 3.4% of H-AI reviews as "AI". Breaking this down by the LLM used for polishing, Pangram's FPR on H-AI reviews is 0.5% (GPT-5), 5.3% (Gemini-2.5-pro), 1.3% (Gemma-3-27b-it), 3.2% (Llama-3.1-70B), and 5.5% (Qwen-3-30B). For GPTZero, the corresponding rates are 1.3% (GPT-5), 2.9% (Gemini-2.5-pro), 1.8% (Gemma-3-27b-it), 1.8% (Llama-3.1-70B), and 9.2% (Qwen-3-30B).

## C. Additional H-AI variants

Prompting general-purpose LLMs represents one popular mode of AI-based assistance in polishing review drafts. We acknowledge the real life polishing practices with AI-assistance can be more diverse. Broadly, we intend H-AI to serve as a blanket category for all forms of AI usage allowed by a polishing-only policy. **Grammarly** naturally fits the polishing-only criterion. Although conferences don't take an explicit stand on **reviews translated with AI-based tools**, such usage should be permitted under polishing-only policy assuming these tools don't tamper with the technical content significantly. Additionally, we consider the scenario where a reviewer supplies **personalized stylistic instructions** to the LLM during polishing so that the polished output better reflects their natural writing style. We conduct experiments to investigate these forms of AI-based polishing.

**Grammarly.** For 100 randomly sampled human reviews from our dataset we manually accept all suggestions (representing maximal modification) in the Grammarly Proofreader web interface. Pangram classifies 89 of these edited reviews as "Human" and the rest as "Mixed" (none are clas-

*Table 9.* Percentage of back translated reviews Pangram classifies as fully "AI"-generated

| Forward Translator | Backward Translator | de | zh | ja |
|---|---|---|---|---|
| Google Translate | DeepL | 1.9 | 1.1 | 3.4 |
| DeepL | Google Translate | 1.5 | 4.5 | 1.1 |

sified as "AI"). This small-scale result demonstrates that under polishing-only policy, Grammarly-style edits on human reviews do not pose the risk of false positives under Pangram, and therefore polishing-only policy may be enforced for Grammarly. A larger-scale study, due to absence of API, requires more human effort and we leave that to future work.

**LLM-based Translation.** To the best of our knowledge, no open-source dataset of non-English peer reviews is publicly available. As a proxy, we translate human-written English reviews into German (de), Chinese (zh), and Japanese (ja), and treat each translation as a stand-in for a review originally written in that language. These reviews are then translated back to English. Reusing a single translator in both directions risks artificially preserving the original phrasing and biasing the evaluation. Therefore, we use different translators, namely Google Cloud Translate[6] and DeepL[7], for the forward and backward directions. Table 9 reports that a non-trivial percentage of these reviews are classified as fully "AI" generated by Pangram.

**Personalized polishing instructions.** While asking an LLM to polish their review, reviewers can add stylistic instructions, so that the polished output better reflects their idiosyncratic writing style rather than generic LLM phrasing. To simulate this, we generate reviews with existing H-AI prompts augmented with additional stylistic instructions (specifically, "use British English", "use short sentences", and "do not use semicolons and em-dashes"). Pangram classifies $2.42\%$ of these reviews as "AI" generated.

While Grammarly-based edits appear safe from Pangram-based flagging, the other two forms of polished reviews yield FPRs high enough to keep our unenforceability claim for polishing-only policies intact. Even if we take $1\%$ (the lowest rate observed across our translation experiments) as a conservative lower bound on the FPR across all H-AI variants, hundreds of policy-compliant reviews would still risk being flagged at the scale of a venue like NeurIPS. We note, however, that real-world polishing behaviour is private and largely unobservable in practice. While we aim to be comprehensive, the variants studied here are simply representative of plausible usage rather than an exhaustive enumeration.

---

[6]https://cloud.google.com/translate
[7]https://developers.deepl.com/

## D. Off-the-Shelf Detectors

### D.1. Descriptions

In this section, we detail the detectors used as automatic benchmarks in Table 2 We evaluate a mix of commercial closed-source and open-source baselines to provide comprehensive coverage of detection approaches.

- **LogLikelihood** (Solaiman et al., 2019) is the simplest detection method where we compute the average log-likelihood of the text under a given language model. Higher likelihoods indicate AI-generated text, while lower likelihoods suggest human-written text. Our LogLikelihood detector uses LLaMA-3-8B for scoring.

- **Binoculars** (Hans et al., 2024) is an open-source detector that works by contrasting token log-probabilities from an observer language model with those from a performer model to compute a cross-entropy difference signal. We use Falcon-7B as the observer models and Falcon-7B-Instruct as the performer model.

- **Fast-DetectGPT** (Bao et al., 2024) is an open-source detector that improves upon the standard DetectGPT by replacing expensive perturbation-based detection with efficient likelihood-based scoring. It operates by comparing token-level log-likelihoods from a base language model against alternative token choices from a reference model, computing a normalized discrepancy score across the sequence. We use Falcon-7B for scoring and Falcon-7B-Instruct for sampling.

- **GPTZero** (Tian & Cui, 2026) is a commercial closed-source detector that uses a deep neural classifier, trained end-to-end on large corpora of human and LLM-generated text from multiple models. As a proprietary system, specific architectural and training details are not publicly disclosed. We use the GPTZero API dated 03.01.2026. We use argmax decoding to choose the highest probability class among "ai", "mixed" and "human" for the final prediction.

- **Pangram** (Emi & Spero, 2024; Thai et al., 2025) is a proprietary detector that uses a transformer-based neural network trained using negative mining with synthetic mirror prompting with human samples and LLM generated samples that closely match them. In our work, we access the latest Pangram version 3.0 dated 03.01.26, which is reportedly based on their recent work on quantifying AI editing (Thai, 2025). We use the labels returned in the "prediction_short" field.

### D.2. Pangram and GPTZero: Easy subset results

Figure 5 presents the distribution of easy subset reviews over the three outcome categories ("AI," "Mixed," and "Human")

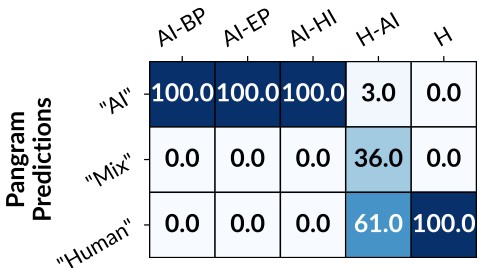 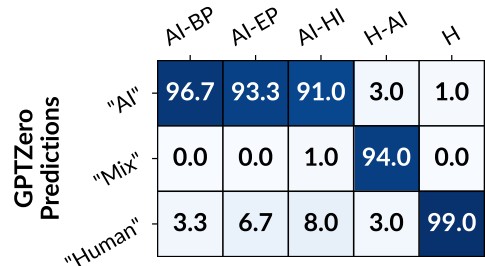

*Figure 5.* Confusion matrices denoting % of reviews classified as "AI", "Mixed", "Human" by Pangram & GPTZero on easy subset.

as predicted by Pangram and GPTZero.

# E. Detection with Peer-Review-specific Advantages

## E.1. Context-aware versus Context-agnostic Zero-shot Detection: Statistical Significance Tests

To test the statistical significance of our results, we use McNemar's test. McNemar's test is a non-parametric statistical test used to compare whether two related classifiers differ significantly in their predictions on the same set of instances (McNemar, 1947). The test is based on a $2 \times 2$ contingency table of paired outcomes and considers only the discordant pairs. Let $b$ denote the number of instances misclassified by the first method but correctly classified by the second, and $c$ denote the number of instances correctly classified by the first method but misclassified by the second. The null hypothesis states that neither of the two models performs better than the other.

For sufficiently large numbers of discordant pairs, $(b+c) \geq 25$, we use the continuity corrected McNemar version (Edwards, 1948), where the McNemar test statistic is computed as,

$$\chi^2 = \frac{(|b-c|-1)^2}{b+c},$$
$$p = P(\chi^2_\nu \geq \chi^2)$$

When the total number of discordant pairs is small, $(b+c) < 25$, we use the exact McNemar test, which models $b$ as a binomial random variable under the null hypothesis and compute a two-sided p-value as,

$$b \sim \text{Binomial}(b+c, 0.5),$$
$$p = 2 \cdot \min\left[P(B \leq b),\ P(B \geq b)\right].$$

In the special case where $b = c = 0$, the p-value is reported as 1, indicating perfect agreement between the paired methods. All McNemar statistics and p-values are computed using the `statsmodels` Python library (Seabold & Perktold, 2010).

*Table 10.* P-values from McNemar's test comparing detectors with their context-aware versions.

| Detector Group | p-value | | | |
|---|---|---|---|---|
| | AI-BP | AI-EP | AI-HI | H-AI |
| Split: **Base** | | | | |
| LogLikelihood | 0.37 | $7.0e^{-3}$ | $1.5e^{-39}$ | $3.0e^{-3}$ |
| FastDetect | 1 | 0 | $4.1e^{-4}$ | $3.5e^{-11}$ |
| Binoculars | 1 | 0.2 | $5.3e^{-6}$ | 1 |
| Split: **Hard** | | | | |
| LogLikelihood | $1.6e^{-22}$ | $4.0e^{-13}$ | $2.3e^{-58}$ | $1.7e^{-18}$ |
| FastDetect | $3.1e^{-5}$ | $4.8e^{-7}$ | $9.9e^{-13}$ | $2.8e^{-19}$ |
| Binoculars | 0.11 | $7.0e^{-3}$ | 0.03 | 0.62 |

## E.2. Detection with Reference AI Reviews: Implementation Details

In matching-based similarity metrics (Soft n-gram matching and Soft keypoint matching), we consider there is a *close match* between two semantic units (n-grams or keypoints) if the cosine similarity between their vector embeddings is greater than a predefined threshold $\tau$. The final similarity metric is the fraction of semantic units in the candidate review that have a *close match* with at least some semantic unit in any of the references. For Soft n-gram matching, we use $n = 40$ (a choice motivated by two competing factors— $n$ should be large enough to capture meaningful semantic information, but not so large that its vector embedding only fits coarse topical information). For Soft keypoint matching, key ideas are extracted using GPT-5-mini with a prompt that asks it to extract important points from the review text. We experiment with the following pretrained embeddings:

**Linq-Embed-Mistral.** This is a general-purpose text-embedding model (Choi et al., 2024) trained for retrieval tasks, which produces 4096-dimensional embeddings. We load it from the publicly available huggigface checkpoint.[8]

**text-embedding-3-small.** We use OpenAI's general purpose text-embedding model, which produces 1536-

---

[8]`https://huggingface.co/Linq-AI-Research/Linq-Embed-Mistral`

*Table 11.* Classifier trained on similarity to AI generated references as features. High error rates across-the-board indicate infeasibility of individual review level predictions with these features.

| Similarity metric | TPR (%) ↑ | | | FPR (%) ↓ | |
|---|---|---|---|---|---|
| | AI-BP | AI-EP | AI-HI | H-AI | H |
| **specter2** | | | | | |
| Cosine similarity | 93.9 | 94.3 | 88.4 | 12.4 | 5.0 |
| Soft n-gram match | 95.7 | 95.7 | 89.8 | 28.0 | 21.0 |
| Soft keypoint match | 94.0 | 90.3 | 88.5 | 15.9 | 11.0 |
| **text-embedding-3-small** | | | | | |
| Cosine similarity | 99.1 | 98.7 | 91.9 | 7.4 | 2.0 |
| Soft n-gram match | 96.9 | 96.8 | 91.1 | 6.3 | 4.0 |
| Soft keypoint match | 92.8 | 88.8 | 80.8 | 4.9 | 2.0 |

dimensional embeddings, accessed via their API,[9] on January 11, 2026.

**specter2.** Released by AI2 and designed for scientific text (Singh et al., 2022), specter2 produces embeddings of size 768 and we load the publicly available huggingface checkpoint.[10]

Following Thai et al. (2025), the results reported in the main paper (Table 4, Figure 3) are based on Linq-Embed-Mistral as the embedding model. The other embedding models also lead to similar outcomes and the respective numbers are reported in Table 11.

### E.3. Details of Stylometric Features to Characterize Peer Review Writing Style

In this section, we describe the 38 stylometric features used to characterize peer-review writing style and used to train a supervised XGBoost classifier. These features capture lexical diversity, syntactic structure, grammatical composition, sentiment and emotion, and readability. For part-of-speech tagging, we use the `pos_tag` function of the `nltk` python library.[11] For syllable-related features, we use the `pyphen` python library to hyphenate words and use that as a proxy for syllables.[12] For emotion-related features, we define emotion seed words grouped into positive (e.g., good, happy, joy, love), negative (e.g., sadness, anger, fear, guilt), and other emotions (e.g., surprise, empathy). Using pre-trained GloVe 100-dimensional embeddings (Pennington et al., 2014), words are assigned to emotion categories based on cosine similarity to the corresponding seed words in the embedding space.

---

[9] https://api.openai.com/v1/embeddings

[10] https://huggingface.co/allenai/specter2

[11] NLTK: Natural Language Toolkit library (v3.9.2), https://www.nltk.org/, accessed December 2025.

[12] Pyphen: Python hyphenation library (v0.17.2), https://pypi.org/project/pyphen/, accessed December 2025.

1. **Average Word Length** - Ratio of number of non-space characters to number of words.

2. **Average Sentence Length** - Ratio of number of words to number of sentences

3. **Type Token Ratio (TTR)** (Hout & Vermeer, 2007) - TTR is a lexical diversity measured as the ratio of unique words to total words.

4. **Root Type Token Ratio (RTTR)** (Hout & Vermeer, 2007) - RTTR is a length-normalized lexical diversity. It is computed as $\frac{\text{Unique Words}}{\sqrt{\text{Words}}}$

5. **Maas Measure** (Tweedie & Baayen, 1998) Maas is a vocabulary richness measure that is robust to text length. It is computed as $\frac{\ln(\text{Words}) - \ln(\text{Unique Words})}{(\ln(\text{Words}))^2}$

6. **Hapax Legomenon Rate** - HLR is the proportion of words that occur exactly once in the text. It is computed as $\frac{\text{Words occurring exactly once}}{\text{Words}}$

7. **Bigram Uniqueness** - Ratio of unique bigrams to total bigrams.

8. **Trigram Uniqueness** - Ratio of unique trigrams to total trigrams.

9. **Punctuation Percentage** - Percentage of characters that are punctuation characters.

10. **Stop Word Percentage** - Percentage of words that are stop words. For identifying stop words, we use the English stopwords corpus of the `nltk` library.

11. **Question Percentage** - Percentage of sentences that are interrogative (end with ?).

12. **Exclamation Percentage** - Percentage of sentences that are exclamatory (end with !).

13. **Abstract Noun Percentage** - Proportion of total words that are abstract nouns.

14. **Sparse Abstract Noun Percentage** - Sparse abstract nouns are low-frequency abstract nouns that are not among the top 5,000 words in the Brown corpus (Kučera & Francis, 1967). For this feature, we use the percentage of abstract nouns that are sparse abstract nouns.

15. **Verb Percentage** - Percentage of total words that are verbs.

16. **Sparse Verb Percentage** Sparse verbs are low-frequency verbs that are not among the top 5,000 words in the Brown corpus (Kučera & Francis, 1967). For this feature, we use the percentage of verbs that are sparse verbs.

17. **Adjective Percentage** - Percentage of words that are adjectives.

18. **Sparse Adjective Percentage** - Sparse adjectives are low-frequency adjectives that are not among the top 5,000 words in the Brown corpus (Kučera & Francis, 1967). For this feature, we use the percentage of adjectives that are sparse adjectives.

19. **Complex Adjective Percentage** - We define complex adjectives as those that are morphologically complex (e.g., have suffixes such as *-ive*, *-ous*, *-ic*). For this feature, we use the percentage of adjectives that are sparse adjectives.

20. **Adverb Percentage** - Percentage of words that are adverbs.

21. **Sparse Adverb Percentage** - We define sparse adverbs as low-frequency adverbs not among the top 5,000 words in the Brown corpus (Kučera & Francis, 1967). For this feature, we use the percentage of adverbs that are sparse adverbs.

22. **Preposition Percentage** - Percentage of words that are prepositions.

23. **Conjunction Percentage** - Percentage of words that are conjunctions.

24. **Complex Sentence Percentage** - Percentage of sentences containing at least one subordinating conjunction.

25. **Syntax Variety** - Number of unique POS tags in the text.

26. **Emotion Word Percentage** - Percentage of words that are emotion-related.

27. **Positive Emotion Word Percentage** - Percentage of words that are positive-emotion-related.

28. **Negative Emotion Word Percentage** - Percentage of words that are negative-emotion-related.

29. **Other Emotion Word Percentage** - Percentage of words that are other-emotion-related.

30. **First-Person Pronoun Percentage** - Percentage of words that are first-person pronouns.

31. **Second-Person Pronoun Percentage** - Percentage of words that are second-person pronouns.

32. **Polarity** - We use the `textblob` library's sentiment polarity. It uses a Bag-of-Words classifier to obtain the text polarity.[13]

33. **Subjectivity** - Similar to Polarity, we use the `textblob` library's subjectivity score. It uses a Bag-of-Words classifier to obtain the text subjectivity.

34. **VADER Compound Score** (Hutto & Gilbert, 2014) VADER is a lexicon and rule-based sentiment analysis tool. Each word gets a sentiment score, rules are applied according to punctuation, modifiers, negation etc and the final score is normalised. We compute the VADER compound score using the `vaderSentiment` python library.[14]

35. **Average Syllables Per Word** - Average syllable count per word.

36. **Complex Word Percentage** - Percentage of words with at least three syllables.

37. **Flesch Reading Ease** (Flesch, 1948) - Flesch Reading Ease is readability metric based on sentence length and syllable count. It is computed as $206.835 - 1.015 \times \frac{\text{Words}}{\text{Sentences}} - 84.6 \times \frac{\text{Syllables}}{\text{Words}}$

38. **Gunning Fog Index** (Gunning, 1952) - Gunning fog index is a readability metric that estimates the years of formal education required to comprehend a given text. It is computed as $0.4 \times \left( \frac{\text{Words}}{\text{Sentences}} + 100 \times \frac{\text{Complex Words}}{\text{Words}} \right)$

## F. Examples of Human-Written AI-Polished Reviews Flagged as AI

In this section, we present excerpts from H-AI (Human-written, AI-polished) reviews that were predicted as "AI" by Pangram, along with the corresponding sections of their original Human reviews. Despite the high degree of semantic and structural similarity between the two versions, Pangram classifies the H-AI reviews as AI-generated while identifying the Human reviews as human-written.

We use a highlighting scheme that indicates the nature of the differences between the texts. Yellow highlighting denotes changes in vocabulary choice where the underlying ideas and sentence structure remain unchanged. Blue highlighting indicates cases where the content is preserved but the sentence structure or ordering of ideas differs.

EXAMPLE 1

**Human-written review excerpt:** This paper provides an interesting analysis of the conditions which enable generation of natural looking textures. The results is quite

---

[13]TextBlob NLP library (v0.19.0), https://textblob.readthedocs.io/, accessed January 2026.

[14]`vaderSentiment`: Python implementation of the VADER sentiment analysis model (v3.3.2), available at https://github.com/cjhutto/vaderSentiment. Accessed January 2026.

surprising, and analysis is quite thorough. I do think the evaluation methods require more work, but as other reviewers mentioned this could be an interesting line of work moving forwards and does not take too much from this current paper which, I think, should be accepted.

**H-AI (Human-written AI-polished) review excerpt:** This paper presents a thorough and insightful analysis of the conditions required for generating natural-looking textures. The findings are quite surprising, and the analysis is comprehensive.

While the evaluation methods could be strengthened, this limitation does not significantly detract from the paper's core contributions. Further development of the evaluation framework represents a promising direction for future work. Given the novelty of the findings and the quality of the analysis, I believe this paper should be accepted.

EXAMPLE 2

**Human-written review excerpt:** In the paper the authors try to come up with a theory which works for graph clustering in model free framework. However, the result is naive and unconvincing. The authors pretend to construct a new theory but theit theory is just a naive handwaving. Just to give a couple of examples: Theorem 1 (Generic Theorem) ... "which also fits G well is close to C" ... Proposition 3 (Informal) ... "under standard recovery conditions"

**H-AI (Human-written AI-polished) review excerpt:** The authors attempt to develop a theory for graph clustering within a model-free framework. Unfortunately, their result falls short, appearing overly simplistic and unconvincing. Rather than presenting a genuinely novel theory, the authors seem to be relying on unsubstantiated claims. For instance, Theorem 1 (Generic Theorem) contains vague assertions, such as "...which also fits G well is close to C". Similarly, Proposition 3 (Informal) lacks rigor, relying on ambiguous "standard recovery conditions".

# G. Discussion on Policies and Enforcement

## G.1. Current enforcement landscape

Our main analysis examines the question whether current AI text detectors reliably distinguish polishing-only-compliant reviews from policy violations? Enforcement in practice, however, draws on a broader set of mechanisms. We give a brief overview of the current landscape below.

**Trust-based systems and violation reporting.** The predominant and default approach remains trust-based where reviewers are expected to self-certify compliance with their assigned LLM policy and/or disclose their mode of LLM usage (ICML 2025; ICLR 2026). Conferences such as ICML

2025 provide Ethics Violation Reporting forms through which authors and other Program Committee members can report suspected misconduct (ICML 2025).

**Prompt injection (watermarking).** ICML 2026 deployed a novel enforcement mechanism (Rao et al., 2025) based on prompt injection where submission PDFs were watermarked with hidden instructions that, if fed to an LLM, would cause two specific phrases (randomly drawn from a dictionary of 170,000 phrases) to appear in the generated review (Agarwal et al., 2026). Reviewers aware of this attack can possibly get around this by either discovering the watermark or editing the review post generation but this scheme targets the most egregious cases of copy-pasted LLM-generated reviews. It detected 795 reviews ( 1% of all reviews) from 506 unique reviewers who had agreed to the no-LLM policy (Policy A), resulting in the desk rejection of 497 papers authored by reciprocal reviewers who violated the policy. Crucially, every flagged instance was also manually verified by a human.

**General LLM-text detectors for triage.** ICLR 2026 mentioned having used "LLM detection tools" as a first pass filter to flag reviews for potential violations and escalate them to AC and SACs (ICLR 2026 Program Chairs, 2025). The official blog doesn't mention the exact tools they used, but GPTZero has independently claimed to have collaborated with ICLR program chairs for reviewing submissions (Esau et al., 2025).

**Instances of hallucinated references.** Although initial flagging may rely on a combination of ethics violation reporting forms, and AI text detectors used for triage, subsequent action is typically taken only when there is verifiable evidence of LLM misuse, such as hallucinated citations (references to papers, or its authors, that do not exist).

## G.2. Challenges of a No-LLM-use policy

Our findings on the un-enforceability of polishing-only policies naturally point toward the No-LLM-use counterfactual. In this section, we wish to clarify that a No-LLM-use policy is not free of challenges either. To begin with, a complete prohibition on LLM use forgoes the productivity gains that responsible AI assistance could offer reviewers, especially in the face of rapidly growing submission volumes. Beyond this normative concern, the policy also faces operational challenges in enforcement, which we discuss below.

Any detection-based enforcement of a No-LLM-use policy will disproportionately catch careless violators, while pushing more determined ones toward stronger evasion strategies. This effectively incentivizes bad actors to behave more adversarially. In a No-LLM-use policy, the enforcement task is to detect whether any form of AI assistance was used to write the review. Our humanization experiments (Table 3) detectors like Pangram remain reasonably robust to

one-shot automated humanization, flagging ∼93% of humanized AI-BP reviews as either "AI" or "Mixed". However, these experiments only simulate a single pass through an off-the-shelf humanizer. In practice, a determined reviewer can mount more dedicated efforts (e.g. repeated iteration through a detector, manual rewriting, or human-in-the-loop editing) which are plausibly more effective at evading detection than one-shot automated humanization. We leave a quantitative study of such adversaries to future work.

## H. Prompts Used for Review Generation

In this section, we provide the full set of prompts used for generating reviews at each of the four levels of our review dataset. Each level contains four to five prompt variants. The Base split has data generated using Prompt Variant 0 at each level, while the Hard subset has data generated using all prompt variants at each level.

**AI-generated with Basic Prompts (AI-BP)**

Prompt Variant 0:

```
You are a reviewer at an AI
conference.  Write a review of
the given research paper following
the provided reviewer guidelines
in 300-400 words.  Write only the
review.
Conference guidelines-
{GUIDELINES}
Paper-
{PAPER_CONTENT}
```

Prompt Variant 1:

```
You are a reviewer for {CONFERENCE}.
Read the following paper and write
a detailed peer review following
the reviewing guidelines below in no
more than 1000 words.
Conference reviewing guidelines:
{GUIDELINES}
Paper:
{PAPER_CONTENT}
Start your response directly with
the review text.  Do not include any
introductory phrases or disclaimers
(e.g., 'Sure, here is the review').
```

Prompt Variant 2:

```
You are a reviewer for {CONFERENCE}.
In your opinion, how well does the
paper meet the reviewing criteria?
Write your review with explicit
reasoning and justification for your
opinions in no more than 1000 words.
Conference reviewing guidelines:
```

```
{GUIDELINES}
Paper:
{PAPER_CONTENT}
Start your response directly with
the review text.  Do not include any
introductory phrases or disclaimers
(e.g., 'Sure, here is the review').
```

Prompt Variant 3:

```
You are a reviewer for {CONFERENCE}.
You are an expert in the field
relevant to this paper.  Provide
a detailed and critical peer review,
demonstrating deep understanding of
the methodology, related work, and
implications.  Your review should
not exceed 1000 words.
Conference reviewing guidelines:
{GUIDELINES}
Paper:
{PAPER_CONTENT}
Start your response directly with
the review text.  Do not include any
introductory phrases or disclaimers
(e.g., 'Sure, here is the review').
```

Prompt Variant 4:

```
You are an impartial reviewer for
{CONFERENCE}.  Avoid personal
opinions or biases.  Base your
review purely on objective
assessment of clarity, technical
soundness, and novelty.  Your review
should not exceed 1000 words.
Conference reviewing guidelines:
{GUIDELINES}
Paper:
{PAPER_CONTENT}
Start your response directly with
the review text.  Do not include any
introductory phrases or disclaimers
(e.g., 'Sure, here is the review').
```

**AI-generated with Elaborate Prompts (AI-EP)**

For this level, we aggregate the recommended best practices of reviewing from ACL 2017 Last Minute Reviewing Advice, [15], NeurIPS 2020 Reviewer Best Practices, [16], NeurIPS 2025 Reviewer Best Practices, [17], ICML 2025 Tips for Reviewing, [18], and AAAI-26 Guidelines On Writing Helpful Reviews. [19]. Each prompt variant has a set of best practices generated using one or more of these sources.

Prompt Variant 0:

---

[15]ACL 2017 last minute reviewing advice
[16]NeurIPS 2020 Reviewer Guidelines
[17]NeurIPS 2025 Reviewer Guidelines
[18]ICML 2025 Reviewer Instructions
[19]Instructions for AAAI-26 Reviewers

You are a reviewer at an AI conference. Write a review of the given research paper following the provided reviewer guidelines in 300-400 words. Write only the review. Here are some tips and tricks that would help you write a good review.

1. Identify Claims- Clearly outline the main claims of the paper. Look for key phrases like \The contributions of this paper are. . . " to identify them. Conference papers usually have 1-2 claims, while journal articles should have several.

2. Evaluate Support for Claims- Assess how the claims are supported. Prioritize real-world statistically significant experiments, followed by laboratory experiments, demonstrations, simulations, and theoretical analysis (in decreasing order of reliability). Avoid papers with unexplained data or unsupported claims.

3. Assess Usefulness- Determine whether the ideas presented are practically useful. Consider if you or the target audience would use it and why.

4. Check Field Knowledge- Ensure the paper reflects common knowledge in the field. Look for correct use of terms and evidence of understanding of relevant literature

5. Evaluate Novelty- The work should present a significant improvement or innovation over existing approaches. Ensure references are comprehensive, accessible, and relevant, with proper citations.

6. Assess Completeness- Particularly for journal articles, the paper should provide sufficient details for reproducibility. Conference papers may have more limited scope.

7. Discuss Limitations-Check if the paper acknowledges its limitations. Journal articles have more space for this, but it is also important for conference papers.

8. Be Constructive- Suggest solutions to shortcomings rather than just pointing them out.

Focus your criticisms on the paper, not the authors.

9. Avoid Bias- Keep your identity anonymous, and ensure your review is impartial and professional. Avoid comments that could indirectly reveal your identity.

10. Encourage Potential- If the paper has a good idea but is poorly executed, encourage the authors to revise and resubmit

Conference guidelines-
{GUIDELINES}
Paper-
{PAPER_CONTENT}

## Prompt Variant 1:

You are a reviewer for {CONFERENCE}. Write a review of the given research paper following the provided reviewer guidelines. Write only the review. Following are some tips that will help you write a good review:

1. Be thoughtful. The paper you are reviewing may have been written by a first year graduate student who is submitting to a conference for the first time and you don't want to crush their spirits.

2. Be fair. Do not let personal feelings affect your review.

3. Be useful. A good review is useful to all parties involved: authors, other reviewers and AC/SACs. Try to keep your feedback constructive when possible.

4. Be specific. Do not make vague statements in your review, as they are unfairly difficult for authors to address.

5. Be flexible. The authors may address some points you raised in your review during the discussion period. Make an effort to update your understanding of the paper when new information is presented, and revise your review to reflect this.

6. Please avoid biasing your review according to discriminatory criteria not having to do with scientific content or clarity. Please avoid wording that may be perceived as rude or offensive. Although the double-blind review

process reduces the risk of
discrimination, reviews can
inadvertently contain subtle
discrimination, which should
be actively avoided.  Example:
avoid comments regarding English
style or grammar that may
be interpreted as implying
the author is "foreign" or
"non-native".  So, instead of
"Please have your submission
proof-read by a native English
speaker," use a neutral
formulation such as "Please
have your submission proof-read
for English style and grammar
issues."

Your review should not exceed 1000
words.
Conference reviewing guidelines:
{GUIDELINES}
Paper:
{PAPER_CONTENT}
Start your response directly with
the review text.  Do not include any
introductory phrases or disclaimers
(e.g., 'Sure, here is the review').

Prompt Variant 2:

You are a reviewer for {CONFERENCE}.
Write a review of the given research
paper following the provided
reviewer guidelines.  Write
only the review.  Don't go over
1000 words.  Here is some advice
on reviewing:  Read the paper
carefully, critically, and with
empathy.  After reading the paper,
think carefully about whether the
paper has properly substantiated
the claimed contributions.  This may
involve verifying proofs, checking
whether hypotheses are actually
tested by the experiments, checking
whether empirical claims do indeed
follow from empirical results,
etc.  Good judgement is needed
to determine the severity of any
issues that you identify.  It is
helpful to point out minor issues
that are easily fixed, but it is
more important to focus on major
issues that are critical to the main
contributions.  Consider whether
the paper places the research
presented into the context of
current research.  Assessments
about a paper's \originality" and
\significance" often crucially
depend on how the paper compares
to prior works, and thus such prior
works should be cited and discussed
in the paper.  Note that in many

cases, it is difficult and often
unnecessary to cite all related
prior works.  If some relevant prior
works are missed, then think about
whether or not including them would
change the conclusions of the paper.
Some omissions may be considered
minor issues that are easily fixed.
Please give constructive comments
and suggestions to the authors to
help them potentially improve their
paper.  In particular, any comments
about strengths and weaknesses must
be substantiated.
Conference reviewing guidelines:
{GUIDELINES}
Paper:
{PAPER_CONTENT}
Start your response directly with
the review text.  Do not include any
introductory phrases or disclaimers
(e.g., 'Sure, here is the review').

Prompt Variant 3:

You are a reviewer for {CONFERENCE}.
Write a review of the given research
paper following the provided
reviewer guidelines.  Write only
the review.  Don't go over 1000
words.  Here are features that
make a bad review:  A review based
on a very superficial reading of
the paper which asks questions
addressed possibly even in the
introduction.  A review based on
sentiment and ideology rather than
on the merits of the paper.  For
instance, I do not like LSTMs and
thus dislike every paper using them
A review which claims things about
the paper which are unsupported by
evidence.  Statements like \this
has been done previously and is
therefore not novel".  Who did
it, when, and how?  Avoid these
mistakes.  In general, do not use
\I", \you", \the authors", etc., in
your reviews.  Reviews should be
depersonalized as much as possible.
Use terms like \the paper", \the
work", \the project".  The review
should focus on the work and not the
individuals (reviewers or authors).
Avoid referring to yourself.  If
you must refer to yourself, it
should be in the third person
(e.g., \this reviewer") and done
sparingly.  Phrase your comments
as would be appropriate if you were
speaking respectfully to the authors
face-to-face.  Instead of \What
is wrong with this paper?", ask
yourself \How could this paper be

```
better?" When suggesting revisions
in the review, think about whether
the revisions are reasonable
in terms of time and resources
| which of the recommendations
are essential, and which are
nice-to-have but optional?
Conference reviewing guidelines:
{GUIDELINES}
Paper:
{PAPER_CONTENT}
Start your response directly with
the review text. Do not include any
introductory phrases or disclaimers
(e.g., 'Sure, here is the review')
```

**AI-generated with Human Input (AI-HI)**

Key points Generation Prompt:

```
You are given a review written by
a reviewer of an AI conference.
You need to give the key points
discussed in the review as one or
two liners for each point. Write
only the key points.
Review-
{HUMAN_REVIEW}
```

Prompt Variant 0:

```
You are a reviewer at an AI
conference. You are given the
key points to evaluate by your
supervisor, you can elaborate
on that. Write a review of this
research paper following the
provided reviewer guidelines in
300-400 words. Write only the
review.
Key points-
{SUMMARIZED_HUMAN_REVIEW}
Conference guidelines-
{GUIDELINES}
Paper-
{PAPER_CONTENT}
```

Prompt Variant 1:

```
I am a reviewer for {CONFERENCE}.
I have read the paper and jotted
down the following key points. Help
me write the complete reveiw by
elaborating on these key points.
Don't exceed 1000 words and start
your response directly with the
review text. Do not include any
introductory phrases or disclaimers
(e.g., 'Sure, here is the review').
Conference reviewing guidelines:
{GUIDELINES}
```

```
Paper:
{PAPER_CONTENT}
Key points:
{SUMMARIZED_HUMAN_REVIEW}
```

Prompt Variant 2:

```
I am a reviewer for {CONFERENCE}.
I am providing you my initial
impression of the paper in the form
of some pointers I have noted down.
Based on these key points write
the complete review following the
conference reviewing guidelines.
Conference reviewing guidelines:
{GUIDELINES}
Paper:
{PAPER_CONTENT}
Key points:
{SUMMARIZED_HUMAN_REVIEW}
Start your response directly with
the review text. Do not include any
introductory phrases or disclaimers
(e.g., 'Sure, here is the review').
Please don't exceed 1000 words.
```

Prompt Variant 3:

```
Help me write a complete review
for this {CONFERENCE} paper by
elaborating on the following points.
Key points:
{SUMMARIZED_HUMAN_REVIEW}
Conference reviewing guidelines:
{GUIDELINES}
Paper:
{PAPER_CONTENT}
Start your response directly with
the review text. Do not include any
introductory phrases or disclaimers
(e.g., 'Sure, here is the review').
Please don't exceed 1000 words.
```

**Human-written with AI polishing (H-AI)**

Prompt Variant 0:

```
I am a reviewer for a renowned
scientific conference. Help me
paraphrase my review while keeping
the original structure, meaning, and
technical content intact. Write
only the review.
Review:
{HUMAN_REVIEW}
```

Prompt Variant 1:

```
I am a reviewer for {CONFERENCE}.
The following review was written by
```

```
me and I am a non-native English
speaker.  Your task is to improve
the writing for fluency, clarity
and natural phrasing, as a native
speaker would write it.  You must
preserve all technical arguments,
criticisms and core meaning without
alteration.  Your job is only to
polish the review.  The review
should not exceed 1000 words.
Draft review:
{HUMAN_REVIEW}
Start your response directly with
the review text.  Do not include any
introductory phrases or disclaimers
(e.g., 'Sure, here is the review').
```

Prompt Variant 2:

```
I am a reviewer for {CONFERENCE}.
The following is a draft review
written by me.  Help me paraphrase
it to improve grammar and clarity.
You must preserve all technical
arguments,
criticisms and core meaning without
alteration.
Draft review:
{HUMAN_REVIEW}
Start your response directly with
the review text.  Do not include any
introductory phrases or disclaimers
(e.g., 'Sure, here is the review').
```

Prompt Variant 3:

```
I am a reviewer for {CONFERENCE}.
The following is a draft review
written by me.  Help me write a
polished review with improved
grammar and clarity.  You must
preserve all technical arguments,
criticisms and core meaning without
alteration.
Draft review:
{HUMAN_REVIEW}
Start your response directly with
the review text.  Do not include any
introductory phrases or disclaimers
(e.g., 'Sure, here is the review').
```

