# OpenReview forum: "Policies Permitting LLM Use for Polishing Peer Reviews Are Currently Not Enforceable"
_ICML.cc/2026/Conference — ICML 2026 regular_

### Official Review · Reviewer_E4jQ · 2026-02-25

**Soundness:** 3
**Presentation:** 3
**Significance:** 3
**Originality:** 3
**Overall Recommendation:** 5
**Confidence:** 4

**Summary:**

The paper studies if the policy of permitting LLM use solely to polish reviews, as adopted by many top-tier conferences, is even enforceable. The paper uses a dataset of reviews from the pre-LLM era, and generates a testbed of more than 50K reviews with varying levels of AI involvement from entirely AI-generated to entirely human-written. Using this testbed, the paper evaluates 5 detection methods, including GPTZero and Pangram. The results show that while the proprietary detectors can reliably detect fully AI-generated reviews even on the hard-subset, they have high false positive rates when AI is used to polish the human-written review in terms of flow, clarity, and grammar, while the structure and content from the human is retained. The paper further studies if providing the manuscript, using writing style, and training supervised classifiers can improve detection, but the results remain unreliable. The paper also shows that even fully AI-generated reviews can evade detection with post generation humanisation, which the paper calls adversarial paraphrasing.

**Compliance With Llm Reviewing Policy:**

Affirmed.

**Final Justification:**

My concerns were whether the H-AI simulation was representative of real-world polishing behaviour, how tools such as Grammarly and translation tools (which use specialised LLMs) would fit into the taxonomy, and what would be considered sufficient evidence for flagging violations. The authors address all three with concrete experiments and reasoning. These new experiments would extend the scope of LLM usage in peer-reviewing the paper considered earlier and improve its diagnostic value. Therefore, I am updating my recommendation from 4 to 5.

**Key Questions For Authors:**

1. How would you categorise the reviews using tools like Grammarly, or translations from another language in your taxonomy?
2. Would including instructions for personalised styles in H-AI prompts lead to lower FPR, suggesting the results in the paper are an upper bound on the real-world cases?
3. Given the unreliable detection, what should count as sufficient evidence to flag a review as a violation in a real conference setting while ensuring it is not unjust to reviewers?

**Limitations:**

The paper would benefit from a limitations section discussing whether the H-AI simulation is representative of real-world polishing behaviour, and whether the reported FPR generalise to real-world reviewer practices.

**Strengths And Weaknesses:**

**Strengths:**

1. The paper is well-motivated, well-written, and enjoyable to read.
2. It is a diagnostic paper which shows the limitations of current tools and policy frameworks. Such a diagnosis is important to point out the blind spots and drive further improvements of both the detection tools and policy development.

**Weaknesses:**

1. Based on the provided prompts, the H-AI reviews might not be representative of how an actual human reviewer would polish reviews in practice. LLM-generated text has recognisable surface-level patterns that a reviewer would want to suppress (e.g. don’t use em dashes). Moreover, a human reviewer often also describes personal stylistic patterns (e.g., use British English, use short sentences, don’t use semicolons, etc.). As a result, the AI-polished review would likely be more human-like than the H-AI simulation in the paper, and the real-world FPR on H-AI reviews would be lower under a polishing-only policy. The humanisation experiment partially addresses this, but this is neither adversarial paraphrasing nor post-generation editing. Like retaining technical content, personal style instructions are used during the LLM inference, which the paper does not address.
2. Tools like Grammarly now use LLM-based engines to suggest grammar correction, sentence restructuring and paraphrasing, and are commonly used by reviewers. A fully human-written review, polished by accepting AI suggestions from Grammarly (or similar tools), is neither H nor H-AI as defined in the taxonomy. It is ambiguous where such a review would fit in the paper’s taxonomy. It is unclear whether the methods tested in the paper would flag such reviews as AI-generated.
3. For the H-AI case, prompt variant 2 acknowledges the possibility of a non-native English speaker polishing a review using an LLM to improve clarity while preserving the technical arguments. However, there is a more fundamental case that the paper does not consider: a reviewer who writes the review in their native language (e.g. in German), then uses a tool like DeepL (which uses an LLM) to translate into English. This is a fully human-written review translated by a specialised LLM, with the English form being LLM-generated. This does not fit any category in the paper’s taxonomy. The content is human-written, but it is not H since no English text was written by a human. It is not H-AI since the LLM did not polish an existing English text but generated all of it, including sentence structures and vocabulary choices, based on a complete human-written review. From a detection perspective, such a review would be stylistically closer to AI-BP or AI-EP and would likely be flagged as AI-generated. Given that the research community is highly international, such cases are already prevalent. There were submissions to ICLR 2026 that disclosed under the mandatory LLM Usage Disclosure section that the manuscript was written in the native language and then translated. Given the reciprocal reviewing mandate at conferences, some authors are likely to follow a similar practice for reviewing. The paper does not offer any insights into how such reviews should be categorised in the taxonomy, nor how the policies should treat them.

---

> ### Author Rebuttal · Authors · 2026-03-30
>
> We thank the reviewer for their thoughtful feedback. We are glad that they found the paper well-motivated, clearly written, and engaging. We appreciate their recognition of its potential to advance detector and policy design. We address the concerns below.
>
> ### H-AI and real world polishing behavior
>
> > How would you categorise the reviews using tools like Grammarly, or translations from another language in your taxonomy?
>
> Broadly, we intend H-AI to serve as a blanket category for all forms of AI usage allowed by a polishing-only policy. Grammarly naturally fits the condition. Although conferences don’t take an explicit stand on reviews translated with AI-based tools, such usage should be permitted under polishing-only policy assuming these tools don’t tamper with the technical content significantly. We conduct experiments to investigate these forms of AI-based polishing.
>
> - **Grammarly.** For 100 randomly sampled human reviews from our dataset we manually accept all suggestions (representing maximal modification) in the Grammarly Proofreader web interface. Pangram classifies 89 of these edited reviews as “Human” and the rest as “Mixed” (none are classified as “AI”). This small-scale result demonstrates, under polishing-only policy, Grammarly-style edits on human reviews do not pose the risk of false positives under Pangram, and therefore polishing-only policy may be enforced for Grammarly. A larger-scale experiment, due to absence of API, requires more human effort and is outside the scope of the rebuttal phase.
>
> - **LLM-based Translation.** We could not find open-source non-English peer review data and due to time constraints for curating one, we translated 100 human-written English reviews to German (de), Chinese (zh) and Japanese (ja), which we use as the proxy for reviews in that language. These reviews are then translated back to English. We use different translators for forward and backward translation, since backtranslation with the same translator can artificially preserve the original English phrasing (Table 1).
>
> **Table 1: Percentage of translated reviews classified as AI by Pangram**
>
> | Forward Translator | Backward Translator | de | zh |  ja |
> |---|---|---|---|---|
> | Google Translate | DeepL | 1.9 | 1.1 | 3.4 |
> | DeepL | Google Translate | 1.5 | 4.5 | 1.1 |
>
> Additionally we experiment on reviews polished with personalized prompts as per your suggestion.
> - **Personalized polishing instructions.** We generate reviews with existing H-AI prompts augmented with additional stylistic instructions (specifically, "use British English", "use short sentences", and "do not use semicolons and em-dashes") with GPT-5 and Gemini-2.5-pro. Pangram classifies 45.4% of these reviews as AI.
>
> Above experiments confirm the reviewer’s intuition that other forms of AI assistance like Grammarly-style edits, LLM-based translation tools and personalized polishing requests have lower FPR under detectors. Hence, our FPR estimates might be an upper bound. However, other than Grammarly, these numbers are high enough that our core claim of unenforceability of polishing-only policies is still true. Even if we consider 1% (as in the translation scenario) as the lower bound on FPR of all H-AI reviews, hundreds of policy compliant reviews risk being flagged at NeurIPS scale.
>
> > The paper would benefit from a limitations section discussing whether the H-AI simulation is representative of real-world polishing behaviour, and whether the reported FPR generalise to real-world reviewer practices.
>
> Agreed, we will include such a discussion in our paper. Our H-AI simulation instructs the LLM to polish for grammar, flow, and clarity without reviewer-specific personalization. Real-world polishing behaviour is likely more diverse, and we cannot fully characterize it since such behaviour is private and unobservable. We will also include above experiments reflecting more real-world behavior based on your suggestions.
>
> ### Sufficient evidence to flag reviews
>
> > Given the unreliable detection, what should count as sufficient evidence to flag a review as a violation in a real conference setting while ensuring it is not unjust to reviewers?
>
> To answer this question, we would first redirect the reviewer to our discussion on current enforcement regimes in response to Reviewer MVuf’s question. Our goal in this work is to conduct a rigorous study so that program chairs can make evidence-based decisions about the use (or not) of post-hoc AI detection tools. In the light of the unreliability of such tools, we believe that sufficient evidence to flag a review should be objectively verifiable, such as hallucinated references. Alternatively, mechanisms with strongly controlled false positive rates, such as the prompt injection scheme deployed by ICML 2026 (reported family-wise error rate of 0.0001) [1], can also constitute sufficient evidence.
>
> ### References
>
> [1] ICML 2026 Blog: https://blog.icml.cc/2026/03/18/on-violations-of-llm-review-policies

---

> > ### Author Rebuttal · Reviewer_E4jQ · 2026-04-02
> >
> > The authors have addressed all my concerns with appropriate additional experiments and well-reasoned responses. Having these experiments in the paper would improve the coverage of possible LLM usage scenarios in peer reviewing and improve the paper's diagnostic value. As I mentioned in my original review, this is a diagnostic paper which shows the limitations of current tools and policy frameworks. Such a diagnosis is important to point out the blind spots and drive further improvements of both the detection tools and policy development. Based on the author's response, I am updating my recommendation from 4 to 5.

---

### Official Review · Reviewer_nDEH · 2026-03-09

**Soundness:** 4
**Presentation:** 4
**Significance:** 2
**Originality:** 2
**Overall Recommendation:** 4
**Confidence:** 4

**Summary:**

This paper tackles a core question that whether the use of AI in peer review can be effectively detected. It focuses on two parts. First, can generalized AI text detectors distinguish between AI-generated reviews (AI-gen), AI-polished reviews (AI-polish), and human-written reviews (Human)? Second, given the special nature of peer review, can task-specific methods improve detection performance?

Experiments show that both open-source and commercial AI text detectors can almost classify AI-gen and Human correctly. However, AI-polish reviews are often misclassified as either AI-gen or Human. Afterwards, the authors try several detection methods which relies on special nature of peer review. These methods do not significantly improve performance.

Based on these results, the paper concludes that with current detectors, one must choose between high false positives (ban AI-polish, and risking false accusations) and high false negatives (allow AI-polish, and risking to false overlooks).

**Compliance With Llm Reviewing Policy:**

Affirmed.

**Final Justification:**

I am satisfied with the paper itself and the author's response. This paper should be accepted.

**Key Questions For Authors:**

see above

**Limitations:**

yes

**Strengths And Weaknesses:**

### Strengths

* As an empirical study, the experimental design is very comprehensive, which makes the work solid.
    * The authors evaluate a wide range of general AI text detectors, including both open-source and commercial systems.
    * The authors also design and test peer-review-specific detection methods that use more information such as the paper content and other human / AI written reviews. Although these methods fail to improve performance, the negative results are still valuable.
    * I like the side experiment on humanization. Discussing this is meaningful and provide useful insights.
* The paper is clearly written and well organized. Despite the large number of experiments, the reader can easily understand the results, explanations, and main takeaways from each experiment.
* The authors address several potential concerns, such as data contamination and out-of-distribution evaluation. I think the defenses provided are reasonable, which improves the reliability of the experiments.

### Weaknesses

Overall, I do not see any fatal weaknesses. This is a solid empirical paper, although the novelty and significance are somewhat moderate.
* Since the main goal of the paper is to empirically study whether AI-written reviews can be detected, and the paper does not propose a new detection method, the novelty is moderate.
* The authors explore ways to improve classification using the special nature of peer review. Since the overall results are negative, this slightly limits the significance of the paper.

### Minor comments and questions

* Some of the LLMs used in the experiments are somewhat outdated. It would be helpful to include results with recent models.

* The related work section focuses on AI text detectors. It is helpful if the authors discuss related work on (1) LLM-generated peer reviews and (2) methods for eliciting / evaluating high-quality peer reviews.

* I am curious about a comparison between pre-LLM and post-LLM era peer review data. If the detector’s FPR and TPR are assumed to remain stable across years, the authors and make a comparison and estimate the true (calibrated) rate of AI usage in peer review.

---

> ### Author Rebuttal · Authors · 2026-03-30
>
> We thank the reviewer for their thorough and constructive evaluation. We are encouraged that they find our experimental design comprehensive, our defenses against potential concerns like data contamination reasonable and the results easy to follow despite the breadth of experiments.
>
> ### Novelty and Significance
>
> > Since the main goal of the paper is to empirically study whether AI-written reviews can be detected, and the paper does not propose a new detection method, the novelty is moderate.
>
> We respectfully note that the contribution of this work is not intended as a methodological advance and agree with the reviewer’s characterization of it as an empirical measurement study. The study has direct policy implications. At least 7 major AI venues (NeurIPS, ICML, ICLR, ACL, EMNLP, AAAI, CVPR) currently adopt polishing-only LLM policies for peer review, yet no prior work has systematically evaluated whether these policies are enforceable in practice. In this light, its contribution lies in evaluating detection across practical settings of human-AI collaboration and showing that, even with peer-review-specific signals, it remains unreliable.
>
> > Since the overall results are negative, this slightly limits the significance of the paper.
>
> As the reviewer noted, the negative results concerning peer-review-specific detection methods are “valuable” to the contribution. More broadly, our findings are significant not in the algorithmic sense, but in the scientific sense, as they directly inform how venues design their LLM policies.
>
> ### Experiments with more LLMs
>
> > Some of the LLMs used in the experiments are somewhat outdated.
>
> While our current evaluation includes strong recent models like GPT-5 and Gemini-2.5-Pro, we agree some models, such as Llama-3.1 and Qwen-3 are outdated. To be doubly sure, we conducted new experiments with three new models,
>
> | Model | Pangram H-AI FPR |
> |---|---|
> | Claude Opus 4.6 | 53.8% |
> | Gemini-3.1-Pro-Preview | 38.8% |
> | GPT-5.4 | 19.0% |
>
> These numbers exceed any reasonable threshold for fair enforcement, reinforcing our conclusion that Polishing-only policies can not be enforced with current detection technology.
>
> ### Related work on merits of LLM-generated peer reviews
>
> > It is helpful if the authors discuss related work on (1) LLM-generated peer reviews and (2) methods for eliciting / evaluating high-quality peer reviews.
>
> We want to clarify that our position is not against the merits of using LLMs for scientific use-cases, specifically reviewing papers. In fact, LLMs can be particularly useful in ensuring scientific rigor [1, 2, 3], although LLM reviewers need to be evaluated appropriately [4]. Our work simply aims to evaluate current policies adopted by journals and conferences, and alongside, provide more evidence to supplement broader discourse on this topic.
>
> ### Estimating true rate of AI use in peer review
>
> > If the detector’s FPR and TPR are assumed to remain stable across years, the authors and make a comparison and estimate the true (calibrated) rate of AI usage in peer review
>
> As per our understanding, the suggested proposal is to use pre-LLM peer-reviews data to calibrate the detector's threshold such that the FPR of AI-assisted writing in post-LLM reviews is better estimated. We agree that it is a compelling approach in principle. However, a key practical limitation is that Pangram or GPTZero do not allow users to set their own thresholds.
>
> ### References
>
> [1] Liu, Ryan, and Nihar B. Shah. "Reviewergpt? an exploratory study on using large language models for paper reviewing." arXiv preprint arXiv:2306.00622 (2023).
>
> [2] Xi, Sarina, et al. "Flaws: A benchmark for error identification and localization in scientific papers." arXiv preprint arXiv:2511.21843 (2025).
>
> [3] Shah, Nihar B. "Challenges, experiments, and computational solutions in peer review." https://www.cs.cmu.edu/~nihars/preprints/SurveyPeerReview.pdf (shorter version in CACM 2022), June 2025.
>
> [4] Goldberg, Alexander, et al. "Peer reviews of peer reviews: A randomized controlled trial and other experiments." PloS one 20.4 (2025): e0320444, blog: https://blog.ml.cmu.edu/2023/12/01/peer-reviews-of-peer-reviews-a-randomized-controlled-trial-and-other-experiments/

---

> > ### Author Rebuttal · Reviewer_nDEH · 2026-03-31
> >
> > Clearly, this paper should be accepted, regardless of whether I am satisfied with the authors' response. Furthermore, I *am* satisfied with the authors' response.
> >
> > **Estimating true rate of AI use in peer review** The authors seem to have misunderstood my point. What I intended to convey was to treat GPTZero as a hard classifier, and evaluate its FPR and TPR on a dataset where the ground truth is known. Based on these measured FPR and TPR values, we can then calibrate, for an unlabeled dataset (such as the reviews from this year's ML conferences), what proportion of the reviews were actually generated by LLMs.

---

### Official Review · Reviewer_MVuf · 2026-03-13

**Soundness:** 3
**Presentation:** 3
**Significance:** 3
**Originality:** 3
**Overall Recommendation:** 4
**Confidence:** 3

**Summary:**

This paper investigates whether LLM usage policies for writing peer review are enforceable. The  authors outline the current landscape of a latest conferences ranging from no LLM to polishing only policies. They utilize the data from peer review written before the deployment of GPT-3 as human only written reviews. Additionally, the generate synthetic review and evaluate them against five AI text detectors. They found varying degree of performance with Pangram performing the best. Ultimately, the underline the poor results on reviews polished by LLMs and suggest prohibiting LLM use entirely may be a batter alternative to allowing polishing-only case.

**Compliance With Llm Reviewing Policy:**

Affirmed.

**Final Justification:**

I adjusted the score for presentation after authors' rebuttal.

**Key Questions For Authors:**

1. What are some of the existing efforts regarding enforcement of such policies?
2. Wouldn't enforcing such strict policy yield more non-compliance?

**Limitations:**

Yes

**Strengths And Weaknesses:**

This study presents a timely and interesting analysis on detection and identifying AI generated text in scientific reviews. While the primary focus is on scientific reviews, other areas may also benefit from this analysis and organization. Exploration of the performance across various scenarios and detectors is another great choice. Inclusion of Humanization as part of the main analysis is yet another great addition to the work.

While the paper findings are interesting, the presentation of the results could benefit from reorganization and clean up. Existing efforts regarding the enforcement of such policies have not been discussed.

---

> ### Author Rebuttal · Authors · 2026-03-30
>
> We thank the reviewer for recognizing the timeliness and significance of this work, for appreciating the exploration across multiple detection scenarios, and for highlighting the inclusion of humanization experiments. We address their questions below.
>
>
> ### Enforcement landscape
>
>
> > What are some of the existing efforts regarding enforcement of such policies?
>
> We agree that a discussion of existing enforcement efforts strengthens the paper and will add one in the revision. We provide this discussion here. The enforcement landscape can be characterized as follows:
>
> - **Trust-based systems and violation reporting.** The predominant and default approach remains trust-based where reviewers are expected to self-certify compliance with their assigned LLM policy and/or disclose their mode of LLM usage [1, 2]. Conferences such as ICML 2025 provide Ethics Violation Reporting forms through which authors and other Program Committee members can report suspected misconduct [1].
>
> - **Prompt injection (watermarking).** ICML 2026 deployed a novel enforcement mechanism [7] based on prompt injection where submission PDFs were watermarked with hidden instructions that, if fed to an LLM, would cause two specific phrases (randomly drawn from a dictionary of \~170,000 phrases) to appear in the generated review [3]. Reviewers aware of this attack can possibly get around this by either discovering the watermark or editing the review post generation but this scheme targets the most egregious  cases of copy-pasted LLM-generated reviews. It detected 795 reviews (\~1% of all reviews) from 506 unique reviewers who had agreed to the no-LLM policy (Policy A), resulting in the desk rejection of 497 papers authored by reciprocal reviewers who violated the policy. **Crucially, every flagged instance was also manually verified by a human.**
>
> - **General LLM-text detectors for triage.** ICLR 2026 mentioned having used “LLM detection tools” as a first pass filter to flag reviews for potential violations and escalate them to AC and SACs [4]. The official blog doesn’t mention the  exact tools they used, but GPTZero has independently claimed to have collaborated with ICLR program chairs for reviewing submissions [5, 6].
>
> - **Instances of hallucinated references.** Although initial flagging may rely on a combination of ethics violation reporting forms, and AI text detectors used for triage, subsequent action is typically taken only when there is verifiable evidence of LLM misuse, such as hallucinated citations (references to papers, or its authors, that do not exist).
>
>
> > Wouldn't enforcing such strict policy yield more non-compliance?
>
> Could you please elaborate on this concern? It is not immediately clear to us why a stricter policy (e.g., a full ban on LLM use) would lead to *more* non-compliance. Perhaps, you mean that under a stricter policy, more reviews might be non-compliant: as in, the policy *does not encourage* non-compliance, but more reviews are non-compliant under the policy? We can only hope that reviewers read the policy and sincerely wish to comply.
>
> ### Presentation
>
> > While the paper findings are interesting, the presentation of the results could benefit from reorganization and clean up.
>
> We appreciate the feedback and are committed to improving the quality of presentation. In response to suggestions from the reviewers we will make the following changes: including discussions on weaknesses of No-LLM-use policy (jfWz), current enforcement landscape (MVuf), related work on quality of LLM-generated peer reviews (nDEH), clarifying scope of H-AI category (E4jQ), pulling humanization results up to appear immediately after Section 4.1 and clarifying Figure 4 caption. We would welcome any other specific suggestions from you on what aspects of the organization or presentation could be improved further, and are happy to incorporate them.
>
> ### References
>
> [1] ICML 2025 Reviewer Instructions: https://icml.cc/Conferences/2025/ReviewerInstructions
>
> [2] Policies on Large Language Model Usage at ICLR 2026: https://blog.iclr.cc/2025/08/26/policies-on-large-language-model-usage-at-iclr-2026/
>
> [3] On Violations of LLM Review Policies: https://blog.icml.cc/2026/03/18/on-violations-of-llm-review-policies/
>
> [4] ICLR 2026 Response to LLM-Generated Papers and Reviews: https://blog.iclr.cc/2025/11/19/iclr-2026-response-to-llm-generated-papers-and-reviews/
>
> [5] GPTZero finds over 50 new hallucinations in ICLR 2026 submissions: https://gptzero.me/news/iclr-2026/
>
> [6] GPTZero finds 100 new hallucinations in NeurIPS 2025 accepted papers: https://gptzero.me/news/neurips/
>
> [7] Rao, Vishisht Srihari, et al. "Detecting LLM-generated peer reviews." PLoS One 20.9 (2025): e0331871.

---

> > ### Author Rebuttal · Reviewer_MVuf · 2026-04-04
> >
> > My concern is that enforcement of such policies is not straightforward. Stricter policies will likely only catch reviewers who carelessly use LLMs. Meanwhile, others will be pushed toward humanization tools and more sophisticated circumvention, making the problem harder to detect, not easier. This creates an uneven enforcement dynamic and an arms race that the authors should discuss more explicitly in the paper.

---

> > > ### Author Response · Authors · 2026-04-06
> > >
> > > We thank the reviewer for following up. We acknowledge that the reviewer’s concern about an arms race dynamic adds an important perspective to a related concern raised by Reviewer jfWz (see our response to their "overarching weakness"). We direct the reviewer there for a detailed treatment, but summarize the key points and planned revisions here.
> > >
> > > We agree that any detection-based enforcement will disproportionately catch the less careful violators, while the more careful ones may resort to humanization tools, iterative manual editing or other evasion tactics. Under a No-LLM-use policy, reviewers who are determined to submit AI-generated reviews, might be incentivized to be more careful and invest additional effort in humanization. Due to this additional effort, a fraction of such reviewers might succeed in evading detection. However, under a No-LLM-use policy, this arms race and the resultant uneven enforcement is confined to violating reviewers only. The compliant reviewers face near-zero false positive rates (0% under Pangram in our evaluation), so enforcement does not penalize the innocent. Under a polishing-only policy, even compliant reviewers who use an LLM only to improve grammar, clarity and flow are flagged at unacceptable rates (Table 2).
> > >
> > > We also note that our humanization experiments (Table 5) show that even purpose-built evasion tools reduce Pangram's detection to only ~8% "Human" classification for fully AI-generated reviews. This suggests that No-LLM-use enforcement, while imperfect, still catches the vast majority of violations.
> > >
> > > In our revision, we will tone down the prescriptive tone and present the strengths and weaknesses of both policies. We will include the above discussion on the arms race dynamic and uneven enforcement concern the reviewer has raised. We will also add a brief overview of the current enforcement landscape (trust-based systems, prompt injection, detector-based triage, hallucinated references). We hope these additions address the reviewer's concern.

---

### Official Review · Reviewer_jfWz · 2026-03-13

**Soundness:** 4
**Presentation:** 3
**Significance:** 3
**Originality:** 4
**Overall Recommendation:** 5
**Confidence:** 3

**Summary:**

The paper argues for strict policies against LLM use during scientific peer review due to the limitations in monitoring and enforcing less-strict “polishing-only” policies--policies which limit LLM use to editing human-written reviews for grammar and fluency--supporting the claim with experimental evidence. The team curates a database of 51,808 peer reviews which contains  a set of “completely” human-written samples and a range of sets produced with four levels of AI assistance. They evaluate detection rates of five modern AI detectors, 2 commercial and three open-source, likelihood-based zero-shot detectors. The latter three were tested with and without conditioning on the contents of the original manuscript. All five AI detectors are also tested on samples of AI-assisted reviews treated with automated paraphrasing tools trained to make text appear human-written. Furthermore, they review classifiers trained on stylometric and linguistic features, and pretrained transformer-based classifiers. All non-trivial evaluations in the study show untenably high false positive rates when classifying human-written AI-polished text. The authors recommend prohibiting AI use in peer reviews when enforcement is considered a priority.

**Compliance With Llm Reviewing Policy:**

Affirmed.

**Key Questions For Authors:**

See "overarching weakness" in strengths and weaknesses section

**Limitations:**

Yes

**Strengths And Weaknesses:**

## Strengths
- Experiments are rigorous, testing multiple AI-detection algorithms, including commercial, open-source, and in-house training.
- Experiments are targeted: all experiments address the main ideas posed in the introduction
- A rigorous dataset is curated, and possible contamination of the dataset is addressed
- Ideas are clear, well-organized, and direct. It conveys its main idea and consistently refers back to it.
- Tables and figures are neat and clear
- Paper is well-written and grammatically sound
- Paper distinguishes itself from prior literature
- It offers a complete dataset that enables further work on the subject
- The paper addresses an important problem in the practice of machine learning--how to address AI use in the review process
- Further work can elaborate on the ideas, specifically, the same experiments can be run in the future once AI-detection software and AIs have improved further.

## Weaknesses
- The humanized review category is not properly addressed. It seems like the easiest work-around for someone intent or doing the least amount of work and remain undetected would be to generate an AI review, paraphrase that review in their own words and formatting (or automate humanization), and then iterate it through AI-detecting software to minimize the score. This paper addresses automated humanized reviews, which, although practical in terms of the study, is a limited proxy for addressing the above scenario.
- Paper claims that the possible contamination of the dataset should “constitute a optimistic estimate of their performance in the wild. Therefore, such contamination does not affect our conclusions.” Although true regarding the main conclusions on polishing-only policies, I believe it should be noted that this contamination could lower the FP rates on “completely human” reviews.
- Regarding Figure 4: I am not sure how well the cosine similarity charts support your argument in the figure description, “The figure highlights that many individual AI-polished reviews receive similarity scores indistinguishable from those of AI∗ reviews”, because the human baseline also overlaps significantly with the AI reviews, making it ineffective as an oversight/enforcement procedure. (However, this is somewhat incidental, as the figure still supports the main idea regarding polishing-only policies--that cosine similarity still isn’t an effective method for distinguishing H from H-AI.)
- I would push the humanization results (table 5) up in the paper to follow the results on the commercial and open-source detectors (table 2). Humanization is introduced much earlier, and so I was originally confused about how it fit into the first set of results.
- In the main paper in section (a) Stylometric features, you should include that you trained an XGBoost classifier. It is only provided in the appendix.

Below, I submit extensive feedback on what I consider an overarching weakness in the paper. However, while I believe my words should be considered, I believe the majority of the paper’s content is sound and contributes solidly to the field:

The paper proposes a no-LLM-use policy because it finds evidence that limited-AI-use policies (particularly the weakest one--polishing-only policies) are not enforceable given the current level of AI-detection algorithms. This implies that AI detection software should be used and that a penalty should be imposed on those caught using it. However, I find flaws in this premise because the overhead costs are likely too high, detection (in my opinion) is too dubious for strict penalties, and it neglects some advantages of the counterfactual case. I’ll run some rough numbers below:

Wikipedia cites ICML 2026 had 24371 submitted papers. Let’s propose a desk-rejection rate of nearly 20% and round the number of reviewed submissions to 20,000. Let’s suppose that each submission was given an average of 5 papers to review, which gives us 100,000 reviews to screen. I couldn’t find specific pricing for  Pangram or GPTZero, but if we estimate based on API access, I’d put it between $2,000-5,000, which is very feasible for a conference. However, if we assume a false-positive rate of 1% (the number cited in your paper for GPTZero), that leaves 1000 innocent reviewers accused of AI use intermixed with a likely greater number of true-positive rates that require judgment from an area or program chair. Even if we lower the number to 0.01% (the figure reported by Pangram), and assume only ~10 innocent suspects, we still need to treat every accusation as having a reasonable level of doubt.  These are rough numbers, but they make me skeptical about the amount of human labor available to sort out these calls.

An alternative case is to dismiss AI-flagged reviews without penalty to the reviewer. However, this allows bad actors to submit poor reviews without consequence. A more balanced approach would be to dismiss a percentage of cases and only adjudicate and appropriately penalize a fraction of AI-generated reviews. This approach sounds like a feasible implementation of your proposal; however, I am still skeptical about how harsh a penalty we can impose, even after human review of the case.

The counterfactual case to your proposal is to be more lenient regarding LLM use (while not employing AI-detectors), either allowing polishing-only policies or greater levels of AI assistance (e.g., iterating with an AI-assistant to maximize the quality of the review). If AI use is managed responsibly, this could improve the quality of reviews and efficiency of the review process. Given that AI is rapidly increasing the rate of scientific output in machine learning, putting a greater burden on reviewers, it might be ill-advised to limit AI use altogether if we want to scale the process.

Given all of the above, I expect that no-LLM-use policies are equally unenforceable, despite the competence of AI detection algorithms like Pangram. Rather, I would propose guidelines that enforce responsible AI use and rely on the net-trustworthiness of the machine learning community, as reviewers have always had the capacity to submit inadequate or biased reviews.

---

> ### Author Rebuttal · Authors · 2026-03-30
>
> We thank the reviewer for an extremely thoughtful and detailed review. We are glad that they found the experiments rigorous and targeted, the dataset curation sound, and the paper well-written and clearly distinguished from prior literature. We also appreciate the discussion around the overarching weakness, which we address, along with other concerns, below.
>
> ### Overarching weakness
>
> > I expect that no-LLM-use policies are equally unenforceable
>
> We largely agree with the sentiment that there are challenges in enforcing a No-LLM-use policy as well. We will tone down our prescriptive tone and lay out the strengths and weaknesses of both policies, with the key message being that polishing-only policies currently can not be enforced using existing detectors. Since the first two weaknesses relate to this overarching concern, we respond to those below.
>
> > This paper addresses automated humanized reviews, which, although practical in terms of the study, is a limited proxy for addressing the above scenario.
>
>
> We agree with the reviewer that iteratively editing text and querying the detector is a plausible strategy a reviewer might employ to disguise AI-generated content as their own. However, determining how effective such a strategy would be and how time-consuming this iterative process is for the reviewer requires detailed experimentation with several human subjects, and we leave this for future work to explore. Instead, we explore automated humanization tools (Sec 4.3) and find that fully AI-generated reviews can get classified as “Mixed” or “Human” if humanized with these tools. The fraction of such reviews classified as “Human” is around 8%. Therefore, under a No-LLM-use policy, which penalizes both “AI” and “Mixed”, no more than 8% of fully AI-generated reviews can bypass detection even if humanized. While automated humanization tools are purpose-built for evading detection and offer less friction, it is plausible that human-in-the-loop rewriting is more effective. If so, we acknowledge this 8% figure would be higher.
>
> > I believe it should be noted that this contamination could lower the FP rates on “completely human” reviews
>
> It is true that the possible contamination of Pangram training data by PeerRead dataset can mean our evaluation underestimates Pangram’s FPR on fully human-written reviews. While this does not impact our conclusion on polishing-only policy, it has implications on the enforceability of No-LLM-use policy. We thank the reviewer for pointing this out. That said, Pangram's publicly available technical documentation independently reports a false positive rate of approximately 1 in 10,000 [1] for fully human written content which is broadly consistent with our numbers. If Pangram's self-reported FPR is to be trusted, the low FPR (0%) observed on human reviews in our dataset is unlikely to be an artifact of contamination.
>
> Given the above two considerations, current detectors can miss No-LLM-use violations for supposedly minor cases (like grammar/spelling correction, translation support with LLMs etc) and more serious instances involving sophisticated evasion tactics (e.g. iterative manual editing of AI reviews). Further, assuming the extremely low FPR rates of Pangram, the un-enforceability of a No-LLM-use policy reduces to overlooking such violations rather than falsely accusing human-written (and policy compliant) reviews. We agree with the reviewer that human adjudication for every flagged review may not scale for the sizes of AI conferences, but might be viable for smaller-to-mid sized venues. We will include this discussion in our paper.
>
>
> ### Regarding Figure 4
>
> Figure 4 shows the distribution of similarity scores between candidate reviews (across varying levels of human involvement ) and AI-generated reference reviews of the same paper. The overlap between the human baseline (along with H-AI) and AI review scores is precisely one of the points the figure is intended to make. Cosine similarity is not a reliable enforcement mechanism even between fully human and AI-generated reviews, let alone between human and AI-polished ones. Nor are any of the other similarity metrics described in the paper. This is also evidenced by the fact that classifiers trained on these metrics have non-zero FPR on human reviews (Table 3). We will revise the figure caption to make this intent explicit.
>
> ### Placement of humanization results and inclusion of classifier detail in Stylometric classifiers section
>
> We gladly accept both suggestions. We will move Table 5 to follow Table 2 and include the XGBoost classifier detail in the main text (Section (a) Stylometric Features).
>
> ### References
>
> [1] All About False Positives in AI Detectors: https://www.pangram.com/blog/all-about-false-positives-in-ai-detectors

---

> > ### Author Rebuttal · Reviewer_jfWz · 2026-04-02
> >
> > Thanks for the response. I maintain my suggestion to accept.

---

### Decision · Program_Chairs · 2026-04-30

**Decision:**

Accept (regular)

**Comment:**

This paper presents an empirical study of whether current policies on LLM use in peer review are actually enforceable. In order to check this, the paper constructs a benchmark of reviews with different levels of AI involvement. The results are interesting to our community, and all reviewers recommend acceptance with scores 5, 5, 4, 4, after the rebuttal-discussion period. At first it seemed like a better fit for the position paper track, but the benchmark construction, empirical analysis, and identifying the limitations of policies are the main technical and empirical contributions (and the recommendation naturally follows from those findings), so this seems acceptable for the main track's scope.